# Exploring the Heterogeneity of Cancer-Associated Fibroblasts via Development of Patient-Derived Cell Culture of Breast Cancer

**DOI:** 10.3390/ijms26167789

**Published:** 2025-08-12

**Authors:** Anna Ilyina, Anastasia Leonteva, Ekaterina Berezutskaya, Maria Abdurakhmanova, Mikhail Ermakov, Sergey Mishinov, Elena Kuligina, Sergey Vladimirov, Maria Bogachek, Vladimir Richter, Anna Nushtaeva

**Affiliations:** 1Scientific Center of Genetics and Life Sciences, Sirius University of Science and Technology, 1 Olimpiysky Avenue, Sirius 354340, Russia; ilina.ana@talantiuspeh.ru (A.I.); anastleont@mail.ru (A.L.); vladimrov.sk@talantiuspeh.ru (S.V.); maryambogachek@gmail.com (M.B.); 2Institute of Chemical Biology and Fundamental Medicine, Siberian Branch of the Russian Academy of Sciences, Akad. Lavrentiev Ave. 8, Novosibirsk 630090, Russia; kaitha@berezus.ru (E.B.); m.abdurakhmanova98@gmail.com (M.A.); ermakovm97@gmail.com (M.E.); kuligina@niboch.nsc.ru (E.K.); richter@niboch.nsc.ru (V.R.); 3Department of Natural Sciences, Novosibirsk State University, Pirogova Str. 2, Novosibirsk 630090, Russia; 4Novosibirsk Research Institute of Traumatology and Orthopedics n.a. Ya.L. Tsivyan, Frunze Street 17, Novosibirsk 630091, Russia; smishinov@yandex.ru

**Keywords:** stromal cells, cancer-associated fibroblast, primary cell culture, patient-derived cell cultures, spheroid, heterotypic 3D, hypoxia, fibrosis, breast cancer, tumor microenvironment

## Abstract

Cancer-associated fibroblasts (CAFs) constitute a heterogeneous population of cells within the tumor microenvironment and are associated with cancer development and drug resistance. The absence of a universal classification for CAFs hinders their research and therapeutic targeting. To define CAF phenotypes, we developed patient-derived cell cultures of breast cancer (BC) and validated and characterized four distinct CAF subtypes (S1–S4) by Costa’s classification. Three out of five primary cell cultures of BC demonstrated different functional features rather than fixed cellular states due to the plasticity of the CAF phenotype. CAF crosstalk with cancer cells supported their survival in the presence of anticancer drugs. Based on the analysis of the cytotoxic effect of doxorubicin, cisplatin and tamoxifen, it was demonstrated that CAF-S4 and CAF-S1 cells were sensitive to the action of all drugs investigated, despite the fact that they possessed different mechanisms of action. CAF-S2 cells exhibited the highest level of resistance to the antitumour agents. Homotypic and heterotypic spheroids with CAFs could be used to model the fibrotic area of BC in vitro. The patient-derived cell cultures of CAFs formed spheroids. Hypoxia-activated CAF-S4 have been shown to stimulate the metastatic potential of triple-negative BC cells in a heterotypic spheroid model. Consequently, this study could be a starting point for the development of novel therapeutic strategies that target CAFs and their interactions with cancer cells.

## 1. Introduction

Despite the progress made in the diagnosis and treatment of breast cancer (BC) over the past several decades, it continues to be the second leading cause of cancer deaths among women [1]. The tumor microenvironment (TME) is a heterogeneous system involved in the development of various pathological processes in the tumor [2]. The most prevalent stromal cellular constituents of the TME are fibroblasts, which possess the capacity to proliferate and differentiate into cancer-associated fibroblasts (CAFs). The subsequent interactions between tumor cells and CAFs are known to contribute to progression, invasion and metastasis, as well as to angiogenesis, immunosuppression and the drug resistance of BC cells through the secretion of growth factors, cytokines and chemokines [3].

Tumors have been described as “fibrotic wounds that do not heal,” and chronic fibrosis has been identified as a risk factor for cancer development [4]. Activated fibroblasts in vivo form clusters as a result of the wound healing process, idiopathic pulmonary fibrosis and hypertrophic scars, as well as after radiation therapy [5]. As indicated by the following sources, there is a correlation between the progression of breast tumors and fibrosis-induced increases in ECM stiffness [6,7]. ECM is a non-cellular component of the TME and is primarily composed of the collagen I type. Previous studies have demonstrated that collagen I lines up near the tumor border, thereby facilitating tumor cell migration [8]. It is evident that the TME plays a pivotal role in tumor progression and metastasis. However, critical components of the TME, such as fibroblasts and alterations in the ECM composition, are not incorporated into conventional in vitro assays that investigate tumor cell migration and invasion.

The approaches to studying CAFs can be considered to comprise the following four interconnected strategies [9]: (1) isolation via primary culture or cytokine-induced differentiation; (2) animal models, such as genetically engineered mouse models; (3) in vitro systems, such as 2D/3D co-culture, conditioned medium analysis and organoids, or in vivo co-implantation, to analyze tumor–CAF interactions at cellular and molecular levels; (4) combinations of classical techniques (such as immunohistochemistry and flow cytometry), used alongside emerging tools (such as single-cell RNA sequencing (scRNA-seq) and spatial transcriptomics) to reveal the complex molecular functions of CAFs within the TME. Several studies have demonstrated that CAFs represent a heterogeneous population of activated fibroblasts in the TME, comprising functionally distinct subtypes [10]. Despite their substantial impact on tumor progression, the precise definition of CAFs remains a subject of controversy [11]. There is no general classification system for CAFs, hampering their study and therapeutic targeting [12]. Classifications of CAFs as proposed in several studies are summarized in [13]. Although a wealth of information has been brought by studies attempting to classify CAFs based on scRNA-seq, there are some limitations in these studies. scRNA-seq analyses can be expensive and time-consuming, but they provide a snapshot of the entire transcriptome of individual cells and gene expression patterns. However, they do not provide information on proteins. Though single-cell platforms for spatial transcriptomics have ultra-high throughput, the clustering and detection of CAF transcriptome states may be limited by gene coverage and a lack of key markers [14]. Immunocytochemistry and flow cytometry (FACS) are used to detect and quantify specific proteins on or within cells.

Costa et al. performed a detailed characterization of CAFs of human BC using FACS [15]. Depending on the expression patterns of markers, CAFs have been divided into four subgroups (S1–S4) exhibiting different effects on the tumor. It is hypothesized that CAFs exhibit distinct functional subtypes, defined by molecular markers such as vimentin (Vim), smooth muscle actin alpha (αSMA), fibroblast activation protein (FAPα), fibroblast-specific protein 1 (FSP1/S100A4) and platelet-derived growth factor receptor (PDGFRα) [16]. Due to the absence of distinctive markers, the representation of CAFs is assessed by detecting a combination of different markers. Several studies have identified two primary subtypes: CAF-S1, which manifests a FAP-high phenotype and is associated with adhesion, wound healing and immunosuppression; and CAF-S4, which exhibits a FAP-low and αSMA-high profile and is linked to invasion and metastasis [3]. The CAF-S2 is regarded as negative for all the above-mentioned markers, while the CAF-S3 should be positive for FSP1 and PDGFRβ. CAF-S2 and S3 are considered to be non-activated fibroblasts, and their functions are subject to scientific debate and remain not clearly defined [17,18].

In the current study, a panel of fibroblasts isolated from breast adenocarcinoma tissue and healthy human breast and eyelid skin tissue was obtained. A wide range of molecular markers, which have been previously linked to tumorigenicity, epithelial-mesenchymal transition and stemness, were used in order to confirm fibroblast phenotype. The present study adopted the concept of Costa’s classification to delineate CAF subpopulations. This classification method was supplemented by the incorporation of a marker associated in the scientific literature with fibroblast inflammatory phenotype (CD90) [19]. The following aspects require elucidation: (i) CAF subpopulations to which the isolated fibroblasts belong; (ii) the correlation between certain CAF subpopulations and sensitivity to chemopreventive agents (tamoxifen, doxorubicin and cisplatin); (iii) the feasibility of culturing certain CAF subpopulations in 3D; (iv) the impact of oxygen deficiency on the mobility of CAFs in a heterotypic spheroidal model.

## 2. Results

### 2.1. Characterization of a Panel of Patient-Derived Fibroblast Cell Cultures

The isolation of cells from tissue samples and the establishment of new cell cultures using standardized protocols have emerged as pivotal instruments in biomedical research. This study utilized the established protocol for the acquisition of patient-derived mammary gland cell cultures, which had undergone previous development and modification [20]. The characteristics of the biological material obtained for the patient-derived cultures are presented in Table 1. The BrC4f patient-derived cell culture had already been defined as a CAF cell culture [20,21,22].

The application of hematoxylin and eosin staining to cells from primary cultures revealed the heterogeneity of the cellular composition of each culture, as well as the variation in the ratios of cells from different populations between patient-derived cultures (Appendix A). Patient-derived cell populations showed a typical spindle-shaped cell morphology.

Cell proliferation rate was assessed using the iCELLigence Real Time Cell Analyzer (RTCA) system, which makes it possible study the growth dynamics of adherent cell cultures in real time on the E-plate system [23,24]. The cell index assesses the characteristics of cell growth kinetics: (1) initial adhesion, (2) proliferation, and (3) cell viability; these can vary between cell cultures (Figure 1a). Cell index (CI) is a parameter reflecting the impedance of electron flow caused by adherent cells. When cells reach confluence, the CI value reaches a plateau [25]. The increase of CI is a result of cells proliferating, adhering strongly and spreading out on the electrode surface. Strongly adherent cells will have a higher CI than weakly adherent cells, which is specific to stromal cells.

As demonstrated in Figure 1b, the data indicate that the MCF7 cell line (dark red line) and the MDA-MB-231 cell line (red line) demonstrate a rapid adhesion rate, with a doubling time of approximately 40 h. BrC4f cells have been shown to exhibit rapid cell adhesion. Furthermore, the onset of cell proliferation was recorded after 60 h, with a concomitant observation of a long plateau phase. These properties may be related to the seeding density and well coverage employed in the experimental setup. The adhesion cells of BC3f (lilac line), BCf4 (blue line) and Met-Tem (green line) are distinguished by a protracted period of cell adhesion, succeeded by a concise plateau phase. The beginning of cell proliferation was recorded after 20–40 h for Met-Tem and BC4f, and after 50–60 h for BC3f. Normal BN120f (orange line) and NSK1f (grey line) cell cultures exhibit a rapid adhesion rate, and the onset of cell proliferation occurred after 40 h for NSK1f and after 70 h for BN120f. The remaining mesenchymal-like cells are characterized by a prolonged plateau phase associated with cell attachment to the well surface and a delay in the onset of the cell proliferation phase (only after 50–60 h).

As a result, we successfully obtained patient-derived cell cultures that displayed heterogeneity in their cell populations. These characteristics closely resemble the heterogeneity observed in tumor morphology and phenotypes in vivo. Cells are predominantly of the mesenchymal-like phenotype in vitro (Appendix A). The mean cell doubling time is 40 h. No bacterial or mycoplasma contamination was detected in the cells.

### 2.2. Phenotyping of Molecular Markers of Fibroblasts in Patient-Derived Cultures

The precise definition of fibroblasts is still debated. Importantly, no specific markers of fibroblasts have been found so far that are not expressed in other cell types. The fibroblast population was identified using classical mesenchymal cell markers, which were used to confirm the fibroblast nature of the isolated populations, other epithelial or mesenchymal markers, and cancer stem cells [26].

Vimentin is a intermediate filament type III protein and major component of the cytoskeleton in non-epithelial cells, particularly mesenchymal cells [27]. Vimentin and Ki-67 are both cellular markers that can be used to assess cancer progression and prognosis. In the TME, the expression of these markers has been observed not only in CAFs but also in epithelial cells undergoing epithelial–mesenchymal transition, vascular endothelial cells and mesenchymal-derived cells, including adipocytes and myocytes [11]. Immunofluorescence showed that vimentin was detected in cells and visualized the structure of the cell cytoskeleton (Figure 2). Ki-67 expression levels were at or below 50% for patient-derived cells and increased in the series 1% ≤ NSK1f < BN120f < BC4f < Met-Tem ≤ BC3f < BrC4f < BrC1f ≤ 50%≤ MDA-MB-231 ≤ MCF7 ≤ 80% (Figure 2).

CD90, also known as Thy-1, is a widely recognized example of a classic molecule found in fibroblasts [28]. According to the data presented in Figure 3a, three groups of cells with the phenotype can be distinguished: CD90low (Met-Tem), CD90+ (BC3f, BC4f) and CD90high (BrC1f, BrC4f). Moreover, the normal BN120f and NSK1f cultures were CD90+ cells.

Neural (N)-cadherin is usually expressed by mesenchymal, neural, endothelial and poorly differentiated cancer cells, whereas epithelial (E)-cadherin is expressed by epithelial cells and well-differentiated carcinomas [29]. All cells exhibited positive N-cadherin expression and negative E-cadherin expression, with the exception of Met-Tem cell culture (Figure 3b,c). MelCam (CD146) is a stromal cell marker that defines fibroblast subtypes within the hematopoietic stem cell (HSC) niche [30]. Fibroblasts, in contrast to epithelial cells, do not typically express EpCAM. The highest proportion of MelCAM+ cells was found in tumor cell lines and BrC4f. The remaining fibroblast-like cell cultures can be described as negative, and the BC3 cell culture can be described as MelCAMlow. The expression of the EpCAM molecule was only observed in tumor cell lines.

Activation of the epithelial growth factor receptor (EGFR) pathway has been shown to stimulate matrix metalloproteinase (MMP)-14 expression and MMP-2 activation, leading to ECM remodeling [31]. Following analysis of the EGFR expression data, three groups of cells with the following phenotypes can be distinguished: EGFRlow (Met-Tem and BrC1f), EGFR+ (BC3f, BC4f and BrC4f) and EGFRhigh (NSK1 and BN120f) (Figure 3d). The platelet-derived growth factor receptor (PDGFR) has been shown to play a pivotal role in connective tissue remodeling [32]. Two distinct forms of this receptor have been identified: PDGFRα and PDGFRβ. The binding of these proteins to PDGF family members has been demonstrated to initiate the coagulation formation reaction, stimulate angiogenesis, and promote tumor growth and metastasis. The analysis of PDGFRα+ cells enables the classification of cell cultures as either PDGFRα- (BC3f, BC4f, BrC4f, BN120f and NSK1f) or PDGFRα-low (Met-Tem and BrC1f) (Figure 3e).

Differential expression of CD44 and CD24 markers discriminates the epitheliod from the fibroblastoid subset [33]. Stromal cells strongly express CD44 and document a role for stromal CD44 in maintaining stemness. The lack of CD24 expression indicates the absence of epithelial cells in the culture (Figure 3h) [34].

This finding confirms that the culture cells derived from patient tumor samples have a fibroblast-like phenotype, which is consistent with the morphological study data (Appendix A). Characterized cells demonstrated moderate to high expression of vimentin, N-cadherin, CD44 and CD90, confirming that the examined cells were fibroblasts and could be further studied as CAF populations.

### 2.3. Cancer-Associated Fibroblast Cell Surface Markers

Fibroblast-specific protein 1 (FSP-1) is a constituent of the S100 family of small calcium-binding proteins, otherwise designated as S100A4. FSP-1+ CAFs have been shown to promote tumor angiogenesis and cell motility and metastasis by producing ECM proteins and secreting cytokines, including tenascin C, matrix metalloproteinases, and vascular endothelial growth factor-A [35]. FSP1 secretion and association with components of the extracellular matrix (e.g., fibronectin) or neighboring cell receptors will inevitably result in heterogeneous staining around stromal cells [36]. Another marker for CAF identification, F-actin, is an actin-binding protein that serves as a scaffold to connect actin bundles, stress fibers, focal adhesions, Z-discs and subcellular structures, playing an important role in normal cell motility [11]. The expression of F-actin was detected in all cell cultures (Figure 4b). FSP1/S100A4 expression analysis revealed that maximal levels of this protein were detected in BC4f and BrC4f cells (Figure 4a and Appendix A). We hypothesize that this is a biological feature, including the possible presence of a secreted form of FSP1 in BC4f and BrC4f. Other fibroblast-like cells could be characterized as cells with low (BrC1f) or medium levels (BC3f and Met-Tem) of FSP1. The normal fibroblast-like cells, BN120f, were found to be low FSP1+ cells. In contrast to NSK1 cells from the eyelid, normal fibroblast-like BN120f cells exhibited low levels of FSP1+ cells (Appendix A and Figure 3). Fibroblast activation protein (FAP) is a type II membrane protein from the family of membrane-binding serine proteases consisting of α and β subunits. FAP is highly expressed in tumor-associated myofibroblasts and is associated with remodeling of the extracellular matrix [30]. According to the data obtained, FAP expression was characteristic of all BC cell cultures examined (Figure 4c). BC3f and BC4f can be described as highly expressing the marker FAPα+ (Appendix A).

Smooth muscle alpha-actin (α-SMA) belongs to the actin cytoskeletal protein family, and α-SMA is one of the known markers for CAF identification. Activation of the epithelial growth factor receptor (EGFR) pathway stimulates matrix metalloproteinase MMP-14 expression and MMP-2 activation, leading to ECM remodeling [37]. All fibroblast-like cells express different levels of α-SMA (Figure 4d and Appendix A). However, the presence of α-SMA stress fibers represents a hallmark of activated myofibroblasts [38]. BrC4f and Met-Tem were characterized by staining of the entire cell cytoskeletal structure, indicating the formation of stress fibers. Single α-SMA+ cells were found in patient-derived cultures of BrC1f. EGFR expression was characteristic of all cell cultures examined, but the lowest signal level was observed in Met-Tem and BrC1f cells (Figure 4e and Appendix A). Comparison of EGFR representation by flow cytometry showed low content of EGFR+ cells in Met-Tem and BrC1 similar to immunocytochemistry data (Figure 3d). Based on the EGFR expression data obtained, three groups of cells with the following phenotypes could be distinguished: EGFRlow (Met-Tem and BrC1f) and EGFR+/high (BC3f, BC4f and BrC4f) (Appendix A). It is evident that normal cells in culture, designated NSK1 and BN120f, exhibit a high level of expression of the EGFR, as demonstrated in Appendix A.

To summarize, patient-derived fibroblast-like cells were isolated from breast tumors and analyzed for characteristics associated with different phenotypes of CAFs (Table 2). Met-Tem cell culture from metastatic BC into brain tissue was also employed for the classification of CAFs. Fibroblasts derived from normal tissue of breast (BN120f) and eyelid (NSK1f) were utilized exclusively as controls and did not contribute to S1–S4 classification.

### 2.4. Determination of Sensitivity of Breast Tumor Cells to Anticancer Agents

CAF cells play a key role in TME BC and induce resistance to chemotherapy [39]. According to the data obtained, cells of different breast tumor cultures have different sensitivity to the action of antitumor drugs (Table 3). Normal cells of breast (BN120f) and eyelid (NSK1f) tissue were used as control normal cells to compare different types of drugs. With the exception of the BrC1 cell, all CAF culture was sensitive to doxorubicin. Normal tissue samples did not react to doxorubicin, but they were sensitive to tamoxifen. The most sensitive to the action of cisplatin were BrC4f and BC4 cells, and to the action of tamoxifen were BrC4f, BC3f and BC4f cells. At the same time, BC4f and BrC4f cells were sensitive to all three investigated drugs despite their different mechanism of action. The most resistant to the tested drugs were the cells of BrC1f.

### 2.5. Potential for Clustering of Fibroblasts into Spheroids

In vitro cell models play important roles as testbeds for toxicity studies, in drug development, and as replacements in animal experiments. The use of cell aggregations in 3D is a suitable model for cancer research, including the fibrotic zone of the tumor. Co-culture models are typically employed in the study of the fibrotic zone; however, the present study concentrated on the potential for fibroblasts to form monotypic spheroids. Using a previously selected method for culturing spheroidal models from stromal cells, we evaluated the potential of CAFs and normal fibroblasts to form 3D models [22]. As seen in Appendix A, all cells were capable of forming spheroidal models. The cells in the 3D model were viable for 7 days (Figure 5).

### 2.6. Analysis of the Effect of Hypoxia-Activated CAF on Breast Tumor Cells When Co-Cultured in the Tumor-Stromal Spheroid Model

As shown previously, the primary culture of BrC4f cells [21] acquires a more aggressive phenotype under hypoxia via the method of «pulsed hypoxia» [21]. The ratio of the oxygen portion in the medium during «pulsed hypoxia» changes by 7%. Using heterotypic BC models (3D-2), we evaluated the effects of CAF (BrC4f) and hypoxia-activated CAF (BrC4f_Hyp2) on the formation in spheroids. It can be seen that BrC4f and BrC4f_Hyp2 cells are differently distributed in 3D-2: BrC4f cells form distinct clusters within the spheroids, whereas BrC4f_Hyp2 cells are distributed uniformly throughout the spheroid (Figure 6b). Stromal cells were also found to be a contributing factor in the compactization of the spheroidal model [40].

On the obtained heterotypic 3D-2 spheroids, we investigated how CAFs and hypoxia-activated CAFs affect the invasiveness of tumor cells of different types of BC. The data obtained indicate that the co-culturing of tumor cells with BrC4f_Hyp2 cells results in the enhanced release of MDA-MB-231 cells from 3D-2 spheroids (Figure 6b). Co-culturing BrC4f_Hyp2 with tumor cells corresponding to other types of BC (MCF7, SKBR-3) had no effect on invasion compared to the original BrC4f. Since hypoxia-activated CAF enhanced invasion when co-cultured with triple-negative BC cells (MDA-MB-231), it was important for us to evaluate how BrC4f_Hyp2 as part of 3D-2 models affect another metastasis factor, cell motility. According to the data obtained (Figure 6c), the most mobile cells were those in heterotypic spheroids mimicking triple-negative BC with hypoxia-activated CAF (MDA-MB-231/BrC4f_Hyp2). Cells from 3D-2 spheroids of MDA-MB-231/BrC4f_Hyp2 migrated a distance 1.5-fold greater than cells from MDA-MB-231/BrC4f spheroids and 4-fold greater than cells from MDA-MB-231. It can be hypothesized that hypoxia-activated CAFs enhance the metastatic potential in triple-negative BC.

## 3. Discussion

A significant challenge in breast cancer research pertains to the acquisition of clinically pertinent outcomes from in vitro studies of treatment modalities and responses to therapies. These studies frequently employ cellular experimental models [26]. Patient-derived cell culture is a useful model to address the fundamental questions in evolutionary physiology since primary cells retain a «normal» phenotype that recapitulates how the cells would behave in vivo [41]. Traditionally, researchers focus on epithelial tumor cell isolation by eliminating stromal cells from culture when obtaining a new cell line [42,43]. In the present study, we focused conversely on obtaining a panel of stromal cells. Owing to the extensive hyperplasia of connective tissue in the breast tumor microenvironment (TME), CAFs account for 80% of the tumor mass and are the most common stromal cell component in the breast TME [44]. Using a previously developed cell isolation method [20], we successfully isolated and cultured fibroblast-like cells from tissue samples. In addition, fibroblast-like cells were isolated from normal tissue samples (Table 1). As normal fibroblasts exhibit multiple differences compared with cancer fibroblasts in vivo, we first investigated whether these variances remained stable in vitro. The investigation revealed that isolated patient-derived cell culture exhibited a mesenchymal morphology that was analogous to that of the BC cell line MDA-MB-231 (Appendix A). Fibroblasts from normal tissue had a thinner, more elongated and regular spindle-shaped morphology. In contrast, primary fibroblasts from tumor tissue had a large and irregular spindle-shaped or stellate morphology (Appendix A). These results are consistent with several studies showing that tumor and normal fibroblasts exhibit different cell morphology [45]. The doubling time of fibroblasts isolated from cancer samples is shorter when compared to that of fibroblasts from normal tissue (Figure 1). This disparity in proliferation rate is one of the characteristics that can distinguish tumor fibroblasts from normal cells [46]. The proliferation of normal fibroblasts is characterized by a low rate of growth, which is reflected in the low Ki-67 levels observed in the data (Figure 2).

Fibroblast cell markers are considered to be essential molecular signatures, which are pivotal in understanding the identity and behavior of fibroblasts in diverse biological contexts. In contrast to other cell types that are characterized by the presence of highly specific surface markers, fibroblasts do not possess any surface markers that are exclusively their own. The absence of distinctive markers complicates the isolation and identification of these elements [47]. In order to verify the cell type, a detailed analysis was conducted to ascertain the expression levels of various markers associated with the epithelial or mesenchymal phenotype, tumorigenicity, and stem cell characteristics (Figure 2 and Figure 3). CD90 was used as a marker to assess the fibroblast origin of cells [26]. The presence of population heterogeneity was suggested by the finding of CD90-positive (fibroblast-like) and CD90-negative (cancer) cells. The analysis revealed that the prevalence of CD90-positive cells was significant across the majority of cultures, with the exception of a lineage derived from a sample of brain metastasis of BC (Figure 3a). The investigation revealed that the fibroblast-like cells exhibited a high degree of positivity for N-cadherin and a low degree of positivity for E-cadherin and EpCAM (Figure 3b,c,f) [20]. MelCam can directly bind with Wnt1 in fibroblasts, activating fibroblasts via the canonical Wnt/β-catenin pathway. Such interaction is essential for Wnt1-induced fibroblast proliferation and ECM production [48]. Our data show that a high percentage of MelCam-positive cells was characteristic of the BrC4f culture and some positive cells were present in the BC3f culture (Figure 3g). Fibroblasts from the BC microenvironment are known to express the adhesion protein CD44 and are implicated in the development of drug resistance and metastasis. The absence of CD24 expression confirms the absence of epithelial cells in culture [34]. These findings serve to confirm their stromal morphology and population heterogeneity in patient-derived cell culture. CAFs in groups comprise native, normal fibroblasts (low Ki-67), as well as activated, proliferating (Ki67+) or recruited fibroblasts in response to cancer-derived stimuli [49]. Ki67 expression levels may be reduced as a consequence of the activity of the non-proliferative functions of CAF. Given the heterogeneity of the cell cultures obtained, additional analysis of CAF markers was performed to isolate cells into a specific population.

The phenotypic diversity and plasticity of CAFs present both challenges and opportunities for the development of effective cancer therapies. Single-cell RNA sequencing (scRNA-Seq) is a common practice to classify and study heterogeneous subpopulations of CAFs [50]. However, the relationship between transcriptomic and proteomic measurements may not be precise, and therefore additional methods and approaches that investigate the extent to which they match will be useful to clarify the conclusions [51]. CAFs show considerable heterogeneity and are classified into subgroups based on different combinations of biomarkers [52]. Different classifications are used for CAFs in BC. Although some subgroups in different classifications may overlap, the different classifications are not identical to each other.

In our study, we chose as a basis the classification proposed by a group of researchers led by Costa et al. [51]. This classification defines four subtypes of CAFs (S1–S4) that accumulate differently depending on the subtypes of BC. In this study, we supplemented this with other molecular markers to determine cell functionality and did not focus on the molecular subtype of the breast tissue sample for CAF typing (S1–S4). By immunocytochemistry and flow cytometry methods (Figure 3, Figure 4 and Appendix A), the obtained fibroblast cell cultures were classified into specific CAF types, including all four S1–S4 types (Table 2). Despite the existence of well-known CAF markers, such as the α-smooth muscle actin (α-SMA), fibroblast activation protein (FAPα) and fibroblast-specific protein-1 (FSP1/S100A4) that were used for the Costa classification, it is difficult to unambiguously define individual populations without considering the analysis of additional markers. Furthermore, the investigation revealed the presence of plasticity forms of fibroblasts, characterized by the expression of distinct markers. For example, BC3f culture (Vim^+^, α-SMA^low^, FAPα^med^, CD90^+^, PDGFRα^−^, N-Cad^high^, EpCAM^−^) is more likely to be correlated with the CAF-S3 subtype. The plasticity of CAFs refers to their ability to adopt a spectrum of distinct phenotypes or states in response to dynamic changes within the TME. CAF subtypes undergo dynamic transitions during tumor progression and therapy pressure [53]. The low expression of FAPα^med^, α-SMA^low^ and PDGFRα^−^ indicates a model of immunosuppressive fibroblasts in the TME. Based on the marker profile of BC4f cells (Vim^+^, FSP1/S100A4^high^, FAPα^high^, α-SMA^med^, CD90^+^, N-Cad^high^, EpCAM^−^), the CAF-S1 subtype is most likely for these cells. This cell culture correlates with activated myofibroblasts and is representative of the most described type of CAFs, which exhibits immunosuppressive and matrix-modeling functions. At the same time, the high expression of FAPα and low expression of α-SMA in BC3f and BC4f cells may indicate a plasticity of the CAF phenotype. Another variant of plasticity cell culture can be referred to as Met-Tem. On the one hand, the Met-Tem cell (Vim^+^, FSP1/S100A4^med^, FAPα^med^, α-SMA^med^, CD90^+^, PDGFRα^low^, N-Cad^low^, EpCAM^−^) obtained from a brain metastasis tissue sample can be correlated with the CAF-S1 subtype by a combination of markers. However, the low expression of N-Cad and α-SMA may indicate a plasticity state, given the TME in the brain, as well as a rather aggressive type of myofibroblasts (CD90^+^). Since this cell culture was obtained from a metastasis tissue sample, it can be assumed that these cells are characterized by a high metastatic potential. The low expression of key CAF markers for BrC1f cell culture (FAPα^med^, α-SMA^low^, PDGFRα^low^, FSP1/S100A4^low^) allows us to strictly correlate the culture cells with CAF-S2 by the Costa classification. The low expression of FAPα and αSMA in BrC1f is distinct from classical myofibroblasts and may represent an early stage of fibroblast activation or represent a group of stem/progenitor fibroblasts. It should also be noted that BrC1f cells showed resistance to all three studied drugs despite their different mechanism of action (Table 3). Most estrogen-positive tumors were enriched in CAF-S2 cells [15]. Hogstrom et al. showed that isolated CAFs from estrogen-positive tumors play a role in supporting a pro-tumor environment by promoting drug resistance [54]. Moreover, CAFs can induce cancer cells to acquire stem cell-like properties, which are often associated with resistance to therapy [55]. Conversely, the Costa classification system designates fibroblast types CAF-S2 and CAF-S3 as inactivated and associates them with normal fibroblast types. As demonstrated in Table 3, the characterization of cells derived from normal tissue samples as drug-resistant cells is also possible. In contrast, BrC4f cell culture was the most sensitive to the investigated chemotherapeutic agents. BrC4f cell culture (Vim^+^, α-SMA^high^, FAPα^med^, FSP1/S100A4^high^, CD90^+^, PDGFRα^−^, N-Cad^high^, EpCAM^−^, MelCAM^high^) correlated with CAF-S4. High expression of CD90, FSP1 and α-SMA is characteristic of matrix-producing fibroblasts, and low levels of FAPα exclude the immunosuppressive phenotype of fibroblasts. The absence of overexpression of tumor-specific CAF markers for BN120f and NSK1 culture cells describes them as normal fibroblasts (Appendix A and Figure 3). Interestingly, we were also able to detect heterogeneity in S1–S4 CAFs subtypes, including differences in the expression of therapy targets such as EGFR: EGFR^low^ (Met-Tem and BrC1f) and EGFR^+/high^ (BC3f, BC4f, BrC4f). A similar experiment to isolate, characterize and correlate cells by S1–S4 cell types from primary cultures of CAF breast cancer was conducted by researchers Piwocka and colleagues [26,56]. The study authors note that some of the isolated primary cell cultures may have characteristics of both CAF-S1 and CAF-S4 groups.

The formation and presence of fibrotic focus is critical for cancer formation, growth and progression; high fibrosis is closely associated with severe invasiveness and high breast malignancy, leading to early metastasis and poor prognosis [57]. The lack of a realistic in vitro model has hindered the investigation of fibrosis mechanisms and, as the standard, monolayer fibroblast culture has limited applicability. One variant of cellular models is spheroids [58]. Spheroids have emerged as superior in vitro models for studying solid tumors compared to monolayer cultures because of their ability to reproduce key aspects of the TME, including intercellular interactions, ECM deposition, and nutrient and oxygen gradients [59]. Heterotypic spheroids, which include tumor cells and cells in the microenvironment, particularly CAFs and endothelial cells, are currently gaining in popularity more rapidly than other models [60]. More often than not, researchers focus on adding CAFs and investigating their functionality with regard to the effect on tumor cells. In the present study, on the contrary, we evaluated the ability of fibroblasts to form spheroids as a cellular model of fibrosis. All patient-derived cultures of fibroblasts used in this study effectively formed spheroids, and the cells in this model were viable for 7 days without necrotic core formation (Figure 5 and Appendix A). Nishikiori and colleagues successfully established various in vitro 3D spheroid models using CAFs and normal fibroblasts from oral squamous cell carcinomas [61]. To better understand and treat lung fibrosis, Xue and colleagues developed ready-to-use fibroblast spheroids [62]. At the same time, some researchers note that clustering of fibroblasts into spheroids causes a massive proinflammatory, proteolytic and growth response called nemesis, which promotes tumor cell invasiveness and leukemia cell differentiation [63]. There are a small number of papers where spheroids were formed from normal fibroblasts for research. Tan and colleagues created, characterized and optimized spheroids composed of only human fibroblasts, demonstrating increased collagen deposition compared to monolayer fibroblasts. Granto et al. derived mono-spheroids from normal fibroblasts for subsequent study of invasion potentials. In another study, a group of researchers obtained spheroids, from myofibroblasts, to analyze myofibroblast deactivation and showed that the spheroid cells did not trigger apoptosis, necrotic cell death or COX-2 protein induction [64].

Stromal cells comprise a major class of cellular components in the TME and also play a very important role in tumor metabolism, growth, metastasis, immune evasion and treatment resistance. It has previously been shown that BrC4f culture cells acquire an aggressive phenotype under hypoxia [21]. In this study, we determined that BrC4f refers to matrix-producing fibroblasts (CAF-S4) and evaluated the effect of initial CAFs and CAFs activated by hypoxia on the formation of spheroids by BC cells of different phenotypes (Figure 6). According to the data obtained (Figure 6b), BrC4f and BrC4f_Hyp2 are differently distributed in heterotypic 3D-2 spheroids: BrC4f culture cells in co-culture with epithelial cells of all types of BC are characterized by a heterogeneous localization focus, while BrC4f_Hyp2 culture cells are homogeneously distributed. We hypothesize that hypoxia-activated CAFs tend to be more motile as part of heterotypic 3D-2 spheroids and therefore do not form separate isolated clusters. Du et al. showed stable and unstable hypoxia can regulate many mechanobiological characteristics of CAFs and can contribute to transformation of CAFs to assist cancer dissemination and the onset of metastasis [65]. Predominantly, invasion is provided by the activation of synthesis and secretion of matrix metalloproteases, enzymes capable of degrading all types of ECM proteins [66]. Cell motility has a key role in invasion into the basal membrane, migration from the primary focus and further homing in distant organs. Cell movement is enabled by changes in the actin cytoskeleton, polarity and pseudopodia formation. Hypoxia-activated CAFs stimulate the metastatic potential of triple-negative BC cells (MDA-MB-231) when co-cultured in the 3D-2 model (Figure 6c). CAF-S4 accumulates in triple-negative breast cancer to promote metastases [15]. Our findings that hypoxic stress is capable of inducing tumor progression and increasing tumor aggressiveness are consistent with the literature. For example, Comito and colleagues showed that hypoxic stress through hypoxia-induced production of reactive oxygen species, which leads to activation of dermal fibroblasts, increases melanoma aggressiveness. In breast tumors, hypoxia also induces remodeling of the CAF proteome, promoting endothelial sprouting and angiogenesis, leading to tumor progression. Brechbuhl et al. showed that MelCAM+ CAFs produced an environment rich in basement membrane proteins, while MelCAM− CAFs exhibited increases in fibronectin 1, lysyloxidase and tenascin C, all overexpressed in aggressive disease [67]. As demonstrated in the preceding section, BrC4f has been identified as a MelCAM-positive culture (Figure 3g). It is hypothesized that hypoxia and MelCAM may mediate tumor cell migration for triple-negative BC via the regulation of specific secreted factors (such as LOX, MMPs) by CAF-S4. HIF-1α enhances the expression of matrix metalloproteinases (MMP-2 and MMP-9), which are essential enzymes and responsible for breaking down ECM components, including collagen and fibronectin [68]. MelCAM and HIF-1α can interact, and this interaction can influence tumor behavior, including angiogenesis and metastasis [48,69].

In conclusion, cell lines remain a valuable resource for in vitro translational research despite the challenges of long-term preservation [70]. It is important to establish and characterize new CAF lines from malignant patient tumors due to the scarcity of commercially available CAFs. However, a key challenge currently facing CAF biology researchers is nomenclature [71]. Moreover, the issue of CAF heterogeneity raises additional questions; including whether CAF subtypes can interconvert or are more stable and whether their phenotype is maintained or modified upon acquisition and subsequent cultivation.

## 4. Materials and Methods

### 4.1. Primary Cell Culture

The protocol for obtaining primary cultures of mammary gland cells was developed and modified previously [20].

All tissue samples (normal and tumor) were obtained with written, informed consent from patients at the Novosibirsk Municipal Budgetary Healthcare Institution «Municipal Clinical Hospital № 1» (Novosibirsk, Russian Federation), National Novosibirsk Regional Oncology Dispensary and Novosibirsk Research Institute of Traumatology and Orthopedics n.a. Ya.L. Tsivyan (Novosibirsk, Russia) with informed consent of patients. The final diagnosis of cancer was confirmed by hematoxylin and eosin staining of paraffin blocks after surgery (Table 1). For use as a control (non-tumor cells), patient-derived cell culture was obtained from a sample of human eyelid skin tissue obtained during blepharoplasty. The surgical material was provided by the Novosibirsk branch of The S. Fedorov Eye Microsurgery Federal State Institution with the informed consent of the patient (Table 1). The study protocol was approved by the Institute of Molecular Biology and Biophysics SB RAS Ethics Committee (Report#1 from 14 March 2017) in accordance with the World Medical Association Declaration of Helsinki. A fresh tissue sample measuring approximately 1 cm^3^ obtained from the resection was immediately placed in DMEM nutrient medium (Dulbecco’s modified Eagle’s medium) (GIBCO, Life Technologies, Carlsbad, CA, USA) containing antibiotic-antimycotic solution (100 units/mL penicillin, 0.1 mg/mL streptomycin and 0.25 μg/mL amphotericin) (#15140122, Gibco™, Waltham, MA, USA) and transported on ice.

Tissue specimens were then subjected to mechanical dissociation using a scalpel, following which they were transferred to a solution of 20 mg/mL collagenase I (Gibco BRL Co., Invitrogen, Waltham, MA, USA) in DMEM medium. The specimens were then incubated at 37 °C for 15 h on a shaking incubator (Grant Bio, Keison Products, Chelmsford, UK). The samples were then dissociated into individual cells and washed with 10× excess of phosphate-buffered saline (PBS), and the resulting cell suspension was collected by centrifugation at 300× *g*. Cells were plated in IMDM with 10% fetal bovine serum (FBS). In the subsequent passages, cells were cultivated in complete IMDM medium, which was enriched with epithelial cell growth supplement (#6622, Cell Biologics, Chicago, IL, USA), as well as Mito + Serum Extender (BD Biosciences-Discovery Labware). The following substances were used for the cultivation of the sample: 2 mM L-glutamine, 100 U/mL penicillin, 100 μg/mL streptomycin and 250 mg/mL amphotericin B. The cultivation was carried out in four-well plates at 37 °C in a humidified atmosphere containing 5% CO_2_. When 70–80% confluence was reached, the cells were harvested using TrypLE™ (#12604013, GIBCO, Invitrogen, Waltham, MA, USA) solution and sub-cultured for further experimentation. Genetic identification of cell lines was performed using a GOrDIS Plus kit (GORDIZ, Moscow, Russia). The STR profiles correspond to those published in international databases ATCC, DSMZ and Cellosaurus.

### 4.2. Cell Culture

MCF-7 and BrC4 cells were cultured in Iscove’s Modified Dulbecco’s Medium (IMDM) (#I7633-10X1L, Sigma-Aldrich, St. Louis, MO, USA), MDA-MB-231 was cultured in Dulbecco’s Modified Eagle’s Medium (DMEM) (#D6046-1L, Sigma-Aldrich, St. Louis, MO, USA) and A431 was cultured in Dulbecco’s Modified Eagle Medium/Nutrient Mixture F-12 (DMEM/F12) (#42400028, Gibco™, New York, NY, USA). All cultures’ media were supplemented with 10% fetal bovine serum (FBS) (#A316040, Thermo Fisher, Waltham, MA, USA) and with 250 mg/mL amphotericin B and 100 U/mL penicillin/streptomycin (#15140122, Gibco™, Waltham, MA, USA). Cells were cultured at 37 °C with 5% CO_2_ unless other conditions are mentioned.

MCF-7 (#ACC 115, DSMZ, Braunschweig, Germany), MDA-MB-231 (#ACC 65, DSMZ, Braunschweig, Germany) and SK-BR-3 (ATCC, #HTB-30, Manassas, VA, USA) were purchased from the American Type Culture Collection (ATCC, Manassas, VA, USA), and BrC4f and BN120f were obtained in the Laboratory of Biotechnology in the Institute of Chemical Biology and Fundamental Medicine SB RAS (ICBFM, Novosibirsk, Russia) [48]. All cell lines were detected free of mycoplasma contamination with RT-PCR. Genetic identification of cell lines was performed using the GOrDIS Plus kit (GORDIZ, Moscow, Russia). The STR profiles correspond to those published in international databases ATCC, DSMZ and Cellosaurus.

«Pulsed hypoxia» has been used to progressively transform BrC4f cells from mesenchymal to epithelial phenotypes. This method is based on the replacement of the cultivating condition from normoxia to hypoxia with the addition of the conditioned medium. The oxygen content was analyzed using a quadrupole mass spectrometer by the ratio of oxygen to nitrogen (Agilent 6490, Agilent Technologies, Santa Clara, CA, USA). The rating of oxygen to nitrogen was 0.275 ± 0.005 (in control) and 0.256 ± 0.005 (one round of PH). The ratio of oxygen portion in the medium during «pulsed hypoxia» changed by 7%. Cells after the second rounds of «pulsed hypoxia» were named BrC4f_Hyp2 [21].

### 4.3. Hematoxylin and Eosin Staining of Cell Cultures

Cells (1 × 10^4^) growing in four-well culture slides (BD Falcon, Bedford, MA, USA) were washed with PBS and 10% neutral-buffered formalin for 1 h, followed by staining with a hematoxylin–eosin solution (1:7 in 15% ethanol) for 3 min. For dehydration, the preparations were sequentially placed in 70%, 96%, and absolute ethanol for 1 min each and then cleared by incubation in xylene for 3 min.

Cell culture specimens (Appendix A) were visualized using a Nikon Eclipse Ti-S series fluorescence inverted microscope (Nikon, Tokyo, Japan). Image analysis was carried out using the NIS-Elements software (Nikon Instruments Inc., version 5.30.05 64-bit, Melville, NY, USA).

### 4.4. xCELLigence Assay

Cell proliferation and survival were monitored in real-time through the xCELLigence Real Time Cell Analyser (RTCA) system (ASEA Biosciences) by measuring cell-to-electrode responses of the cells seeded in E-plates with the integrated microelectronic sensor arrays (ACEA Biosciences Inc., San Diego, CA, USA). The impedance of gold microelectrodes in RTCA systems when cells are not present or not adhered onto the electrodes is determined with ionic cell culture medium solution. Adherent cells act as an insulator on the surface of the electrode and change the ionic medium of the electrode solution, increasing the impedance [72]. The higher the proliferation and adhesion rate of the cells, the more increased the impedance. The cell index (CI) is a function of the cell number and ratio of cells at different time intervals; CI = 0 when there is no cell adhesion. The CI in a RTCA system is the result of the impedance induced by adherent cells to the electron flow. CI is calculated as follows: CI = (impedance at time point n-impedance in the absence of cells)/nominal impedance value.

Cells were seeded at a density of 2500 cells per well in a total volume of 200 μL of IMDM and were monitored in real time for no less than 110 h. The cell index (CI) was calculated for each E-plate well by RTCA Software 1.2 (Roche Diagnosis, Meylan, France) using 16-well E-plates with integrated microelectronic sensor arrays (ACEA Biosciences Inc., San Diego, CA, USA). The graphs are real-time-generated outputs from the xCELLigence system (Figure 1).

### 4.5. Immunocytochemistry

Cells (1 × 10^4^) growing in four-well culture slides (BD Falcon, Bedford, MA, USA) were washed with PBS and 10% neutral-buffered formalin. To block nonspecific antibody binding, cells were incubated in 1% BSA (Sigma-Aldrich, Burlington, MA, USA) and 0.3 M glycine in PBST buffer (PBS with 0.1% Tween 20) for 30 min at RT. Next, cells were incubated with anti-vimentin (#ab8069, Abcam, Cambridge, UK), anti-Ki-67 (#ab15580, Abcam, Cambridge, UK), anti-S100A4/FSP1 (#MA5-31333, Invitrogen, Waltham, MA, USA), anti-Actin-FITC (F3046, Sigma, USA), anti-FAPα (MAC469Hu22, Cloud-Clone, CCC, Wuhan, China), anti-αSMA (#ab5694, Abcam, Cambridge, UK) and anti-EGFR (#sc-373746, Santa Cruz Biotechnology, Santa Cruz, CA, USA) antibodies for 60 min at RT. For visualization, FITC-conjugated (#ab97050, Abcam, Cambridge, UK) or Alexa Fluor 555-conjugated (#A32727, Invitrogen, Waltham, MA, USA) secondary antibodies were used for 1 h at RT. The results were visualized using a Nikon Eclipse Ti-S series fluorescence inverted microscope. Image analysis was carried out using the NIS-Elements software (Nikon Instruments Inc., version 5.30.05 64-bit, Melville, NY, USA), determining the level of cellular fluorescence (MFI) from fluorescence microscopy images in Fiji software (ImageJ 2.16.0/1/54p, Java 1.8.0_442 (64-bit)). Data are mean ± standard deviation of three replicate measurements performed on the same cells.

#### 4.5.1. Counting the Number of Cells on a Preparation in ImageJ

The image file should then be opened in ImageJ and converted to 8-bit. The threshold must be adjusted, with the sliders being used to highlight the cells so that they stand out from the background. The initial step involves the opening of the analysis particles and the subsequent setting of the parameters. Set parameters: size (pixel^2^): 200–5000; circularity: 0.40–1.00; show: outlines; summarize: exclude on edges. The results of the review are presented in the following section. In order to calculate the intensity level, it is necessary to change the analysis particles and set the parameters: size (pixel^2^): 200–25,000; circularity: 0.40–1.00; show: outlines; summarize: add manager option, exclude on edges. Each cell will be assigned a unique number in the image. It is imperative to note that the ROI Manager window will reflect all cell numbers. The subsequent stage of the process is to open the set measurements and set the parameters. The area and the mean grey value should be entered. The image file should be reopened and measured. The subsequent window will present the counts for all cells, the mean for all cells and the standard deviation.

#### 4.5.2. Counting MFI of the Fluorescent Signal on a Preparation in ImageJ

The image file should then be opened in ImageJ and the channels should be split. The selection of the blue channel is to be followed by adjustment of the threshold. This should result in all nuclei turning red on a black background. The separation of the nucleus is achieved through the implementation of a watershed. The subsequent stage of the process is to select all the nuclei for analysis. The subsequent stage of the process is to save the image with the nucleus as a separate file with a new name for the ROI. The subsequent stage of the process is to select an alternative channel and to create a duplicate of the image. This is a prerequisite for the analysis of MFI in the context of the nuclei and cytoplasm. In the following instance, the duplicated image should be subjected to the following adjustments: the foreground should be rendered in a shade of red with a black background. In the next step of the process, the second image should be selected and the process repeated. The subsequent step involves the opening of the previously saved file with kernels. This process is then repeated for the second image. The subsequent stage of the process is to calculate the signal intensity in the nuclei by selecting the ‘Outside’ option. In order to calculate the signal intensity in the cytoplasm, the second image must be selected, and the ‘Clear’ option must be used. In order to calculate the intensity, it is necessary to select the ‘Set Measurements’ function. In order to do so, the following settings must be configured: area, standard deviation, mean grey value, integrated density, limit to threshold and decimal places—2. To measure the MFI of the nuclei used, use the measuring tool. To measure the MFI of the cytoplasm without the nuclei, first create a selection and then measure.

### 4.6. Flow Cytometry

All analyses were performed using a FACS Canto II flow cytometer (BD Biosciences, Franklin Lakes, NJ, USA), and the data were analyzed using FACSDiva Software Version 6.1.3. (BD Biosciences). Cells were initially gated based on forward scatter versus side scatter to exclude small debris, and ten thousand events from this population were collected. The following antibodies were used for analysis: anti-HER2-FITC, anti-HER3-APC, anti-EpCAM-Percp-cy5.5, Mel-CAM-APC from Sony (San Jose, CA, USA) (#162686 and #162652), anti-CD44-APC and anti-CD24-PECy7 from BD Pharmigen (Franklin Lakes, NJ, USA) (#560890 and #561646, respectively), anti-EGFR antibodies (#MA5-13319; Invitrogen, Rockford, IL, USA), CD90 (#ab134360, Abcam, Cambridge, UK) and anti-PDGFRb (#MA5-15143, Invitrogen, Waltham, MA, USA). Cells were initially gated based on forward versus side scatter to exclude small debris, and ten thousand events from this population were collected. Control cells were treated with appropriate isotype PE-conjugated IgG (BD Biosciences).

### 4.7. MTT Assay

The cytotoxic effects of the cisplatin, doxorubicin and tamoxifen on human tumor cells were investigated using the MTT assay (Sigma-Aldrich; Merck Millipore, Burlington, MA, USA) according to a protocol described previously. The cells that had reached 30% confluence in a 96-well plate were incubated for 48 h with preparations at various concentrations. After incubation, the supernatant was removed, and 200 µL MTT solution in RPMI 1640 medium (0.5 mg/mL) was added to each well and incubated for 4 h at 37 °C. The formazan crystals were dissolved in 150 µL dimethyl sulfoxide. The optical density of the formazan solutions was measured using an Apollo LB912 photometer (Berthold Technologies, Oak Ridge, TN, USA) at a wavelength of 570 nm. Cell viability was determined relative to the viability of the control cells (100%)  ±  standard deviation in three independent experiments.

### 4.8. Spheroid Formation

#### 4.8.1. Stromal Spheroid

Spheroids were from stromal cell multi-well agarose-coated plates. Agarose hydrogel 2% (50 µL) was added to each well of a 96-well culture plate (TPP, Trasadingen, Switzerland) and incubated at 37 °C for 1 h. Stromal cells were seeded in 100 µL growth medium at a concentration of 2500 cells per well. Plates were incubated at 37 °C for an additional 72 h to allow formation of 3D spheroids in culture. Analysis was performed on the results of three independent experiments. The results were visualized using a Nikon Eclipse Ti-S series fluorescence inverted microscope. Image analysis was carried out using the NIS-Elements software (Nikon Instruments Inc., version 5.30.05 64-bit, Melville, NY, USA).

#### 4.8.2. Heterotypic Spheroid

The homo- and heterotypic spheroids of BC were established using the liquid overlay technique in 96-well Nunclon™ Sphera™ U-shaped-bottom plates (#174925, Thermo Scientific, Waltham, MA, USA) [22]. Three immortalized BC lines were used as tumor cells for 3D-2-culture formation: ESR1-positive breast adenocarcinoma MCF-7; HER2-positive breast adenocarcinoma SKBR-3; triple-negative breast adenocarcinoma MDA-MB-231. These were co-cultured with BrC4f and BrC4f_Hyp2. Vital cell staining with CellTracker fluorescent dyes was used to determine cell localization in spheroids. CAFs were stained with CellTracker Green (Invitrogen, Carlsbad, CA, USA), and tumor cells with CellTracker Red CMTPX (Invitrogen, Carlsbad, CA, USA). Analysis was performed on the results of three independent experiments. The results were visualized using a Nikon Eclipse Ti-S series fluorescence inverted microscope. Image analysis was carried out using the NIS-Elements software (Nikon Instruments Inc., version 5.30.05 64-bit, Melville, NY, USA).

### 4.9. Live/Dead Staining

Previously obtained spheroids were stained with 1 µg/mL fluorescein diacetate (FDA) (#F1303, Thermo Fisher, USA) diluted in DMEM/F12 FBS-free medium for 45 min at 37 °C. After washing of FDA with 1× PBS, the spheroids were stained with 20 µg/mL propidium iodide (PI) (BD Biosciences, NJ, USA) and 1:1000 Hoechst 33342 (Invitrogen, USA) diluted in PBS for 10 min at 37 °C. The results were visualized using a Nikon Eclipse Ti-S series fluorescence inverted microscope. Image analysis was carried out using the NIS-Elements software (Nikon Instruments Inc., version 5.30.05 64-bit, Melville, NY, USA).

### 4.10. Investigation of the Potential of Cells in Spheroids for Invasion and Migration

Migration and invasion are two clearly separated terms in experimental cell biology (Appendix A) [73]. Migration is defined as the directed movement of cells on a substrate, which can be a natural hydrogel. In order to assess cell migration abilities in vitro, 2% gelatin can be used as a substrate, as it is a porous, inert material in which tumor cells are able to spread unhindered; thus, this substrate was chosen for our study. Invasion is defined as the movement of a cell through a 3D matrix, which is accompanied by a rearrangement of the 3D environment for cell exit. One of the classical methods to study cell invasion in vitro is the test in substrate from Matrigel^TM^, a gel-like mixture that contains ECM proteins; thus, it was chosen to study the migratory capacity in spheroid models. Cells with invasive potential break down Matrigel^TM^ and form pores in it through which they penetrate deep into the substrate from the spheroids, forming star-shaped protrusions. The bottoms of the wells of a 96-well plate were covered with Matrigel^TM^ hydrogel (BD Biosciences, San Jose, CA, USA) or gelatin from the skin of cold-water fish (Sigma, Oakville, ON, Canada) to form a substrate (Appendix A). The process of polymerization of Matrigel™ hydrogel is as follows: the plate was subjected to a temperature of 37.0 ± 1.0 °C for a duration of 30 min. Then, the obtained spheroids were transferred onto hydrogel substrate for subsequent cultivation under standard conditions. The analysis was conducted at the microscopic level. The invasion or migration of cells from spheroids was performed in two distinct phases: firstly, immediately following the transfer to Matrigel™, and secondly after a 24 h period. The duration of the process is measured in hours. To assess migration potential, we measured the radius of the spheroids immediately after transfer to the gelatin substrate and after 24 h. The difference between the two radius measurements was used to estimate cell motility within 3D-2 spheroids (Appendix A). The results were visualized using a Nikon Eclipse Ti-S series fluorescence inverted microscope. Image analysis was carried out using the NIS-Elements software (Nikon Instruments Inc., version 5.30.05 64-bit, Melville, NY, USA). Analysis was performed on the results of three independent experiments.

### 4.11. Statistical Analysis

Statistical analysis was performed on the results of three independent experiments. Results were analyzed using GraphPad Prism v.9.0 (GraphPad Software, San Diego, CA, USA) and two-way Analysis of Variance (ANOVA). The immunochemistry results obtained are presented as the mean ± standard deviation of three different fields of view.

## 5. Conclusions

Establishing a panel of primary CAF cell cultures capable of preserving and reproducing tumor-specific heterogeneity in vivo represents a practical tool for basic and translational research. Major challenges in CAF research include accurate characterization, identification of individual markers and understanding the unique functions of each group. The established panel of primary CAF cell cultures can be used by commercial companies in the development of new targeting drugs (Table 4).

Consequently, the heterogeneity of CAFs within tumors establishes a multifaceted system, wherein each cellular subtype fulfils a distinct function, thereby enhancing the aggressiveness and protection of the neoplasm. Overall, the study confirms the importance of considering the tumor microenvironment in assessing the efficacy of chemotherapy and opens new avenues for developing more effective strategies for the treatment of breast cancer.

## Figures and Tables

**Figure 1 ijms-26-07789-f001:**
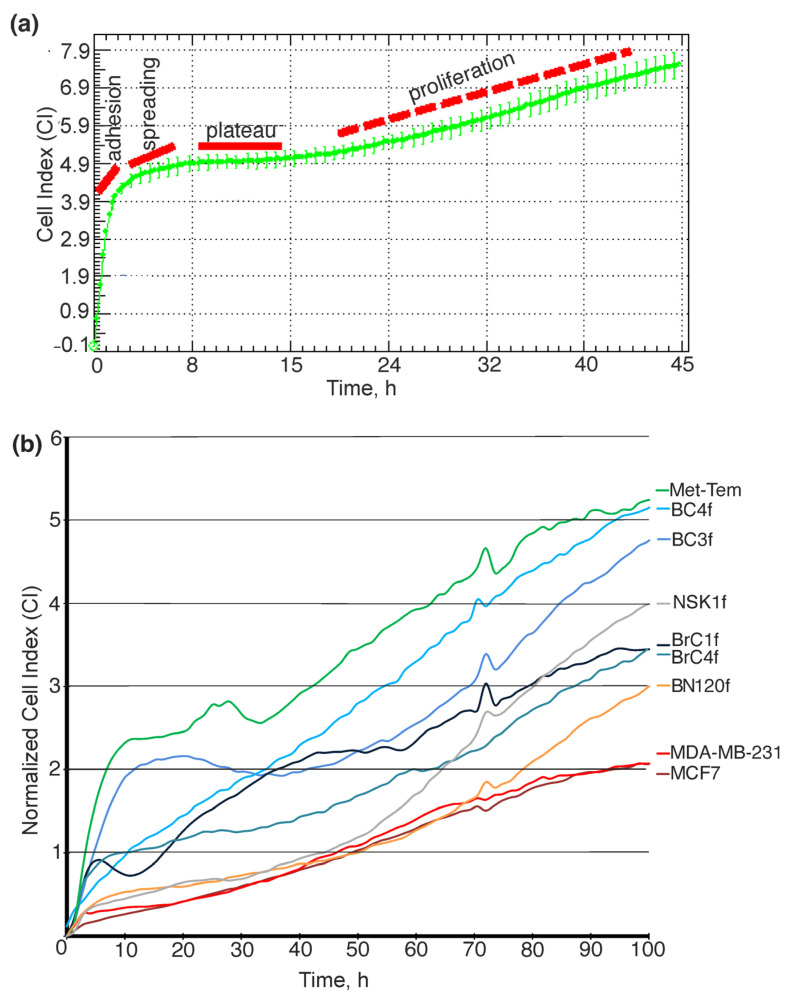
xCELLigence data showing typical Cell Index (CI) curves, which reflect cell proliferation in real-time mode. (**a**) Interpretation of xCELLigence biosensor Cell Index curves: determination of assay parameters influencing experimental design by [23]. (**b**) Direct comparison of growth characteristics of patient-derived culture of breast cancer and normal tissue. The cell cultures used for the study were established using MCF7, MDA-MB231 and BrC4f as control. Data are presented as average of three independent repeats.

**Figure 2 ijms-26-07789-f002:**
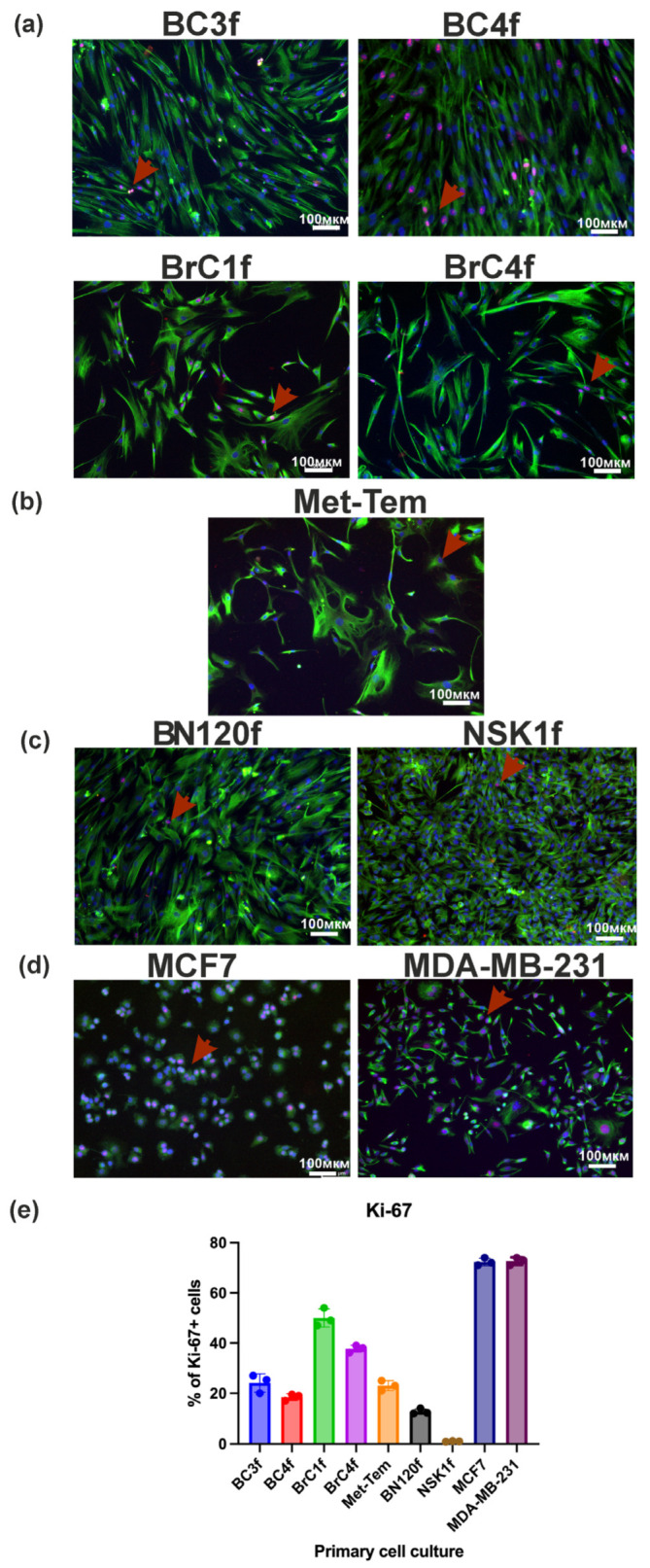
Expression of vimentin and Ki-67 in patient-derived culture cells. (**a**) Fibroblast cell culture from breast tissue; (**b**) fibroblast from metastatic BC brain tissue; (**c**) fibroblast from normal breast tissue BN120f and NSK1 from eyelid tissue; (**d**) epithelial-like MCF-7 and mesenchymal-like MDA-MB-231 BC cell line; (**e**) the percentage of Ki-67+ cells. Triple immunofluorescence staining of vimentin (green), Ki-67 (red) and DAPI nuclei (blue), magnification 10×. Red arrows: Ki-67+ nuclei. Data are mean ± standard deviation of three different fields of view.

**Figure 3 ijms-26-07789-f003:**
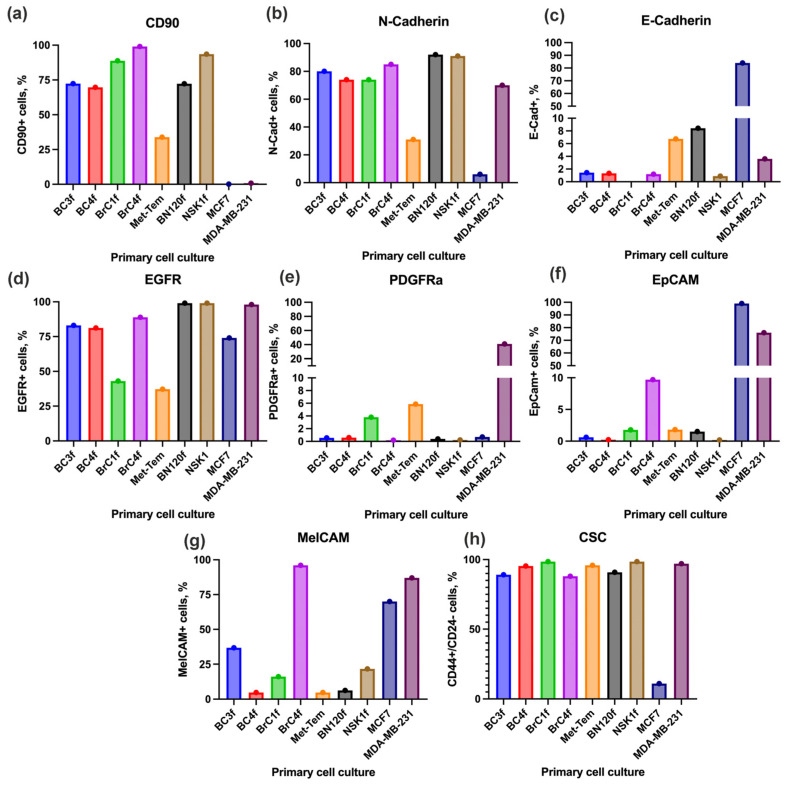
Phenotyping of patient-derived culture cell by flow cytometry. Cells were distinguished using CD90, N/E-cadherin, EGFR, PDGFRa, Ep/Mel-CAM, CD24 and CD44 to confirm their malignant and fibroblast phenotype. Representation of fibroblast marker (**a**) CD90 in cells of patient-derived cultures of BC and normal human eyelid cells. (**b**–**g**) Markers involved in epithelial–mesenchymal transition. (**h**) CSC markers. The analysis was performed by flow cytometry. Immortalized BC line was used as control.

**Figure 4 ijms-26-07789-f004:**
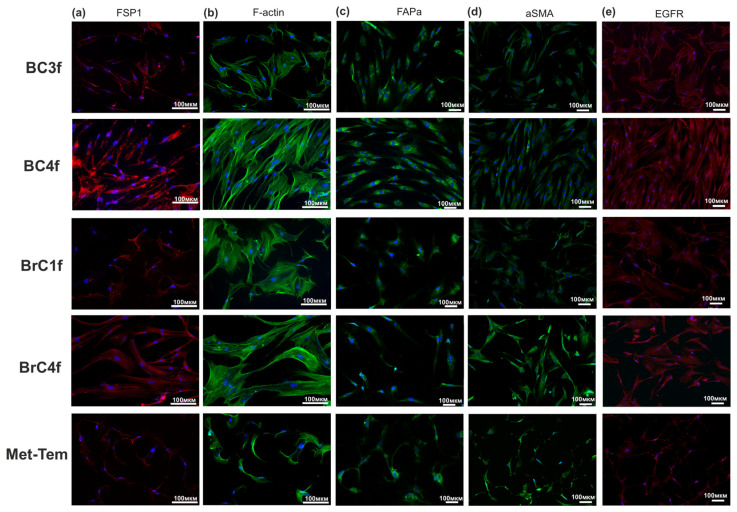
Immunofluorescence staining of markers of cancer-assisted fibroblast. Expression of (**a**) FSP1/S100A4 (red signal); (**b**) F-actin (green signal); (**c**) FAPα (green signal); (**d**) αSMA (green signal); and (**e**) EGFR (red signal) in fibroblast. Hoechst 33342 staining nuclei (blue). Magnification 10×, 20×.

**Figure 5 ijms-26-07789-f005:**
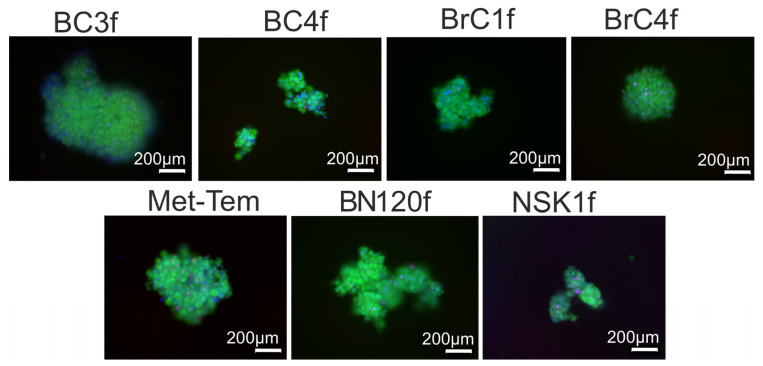
Images of live/dead-stained spheroids after 7 days of culturing. Fluorescence image analysis of the cells in spheroids stained with FDA (green, live cells) and PI (red, dead cells) as well as total cell number (Hoechst 33342/blue).

**Figure 6 ijms-26-07789-f006:**
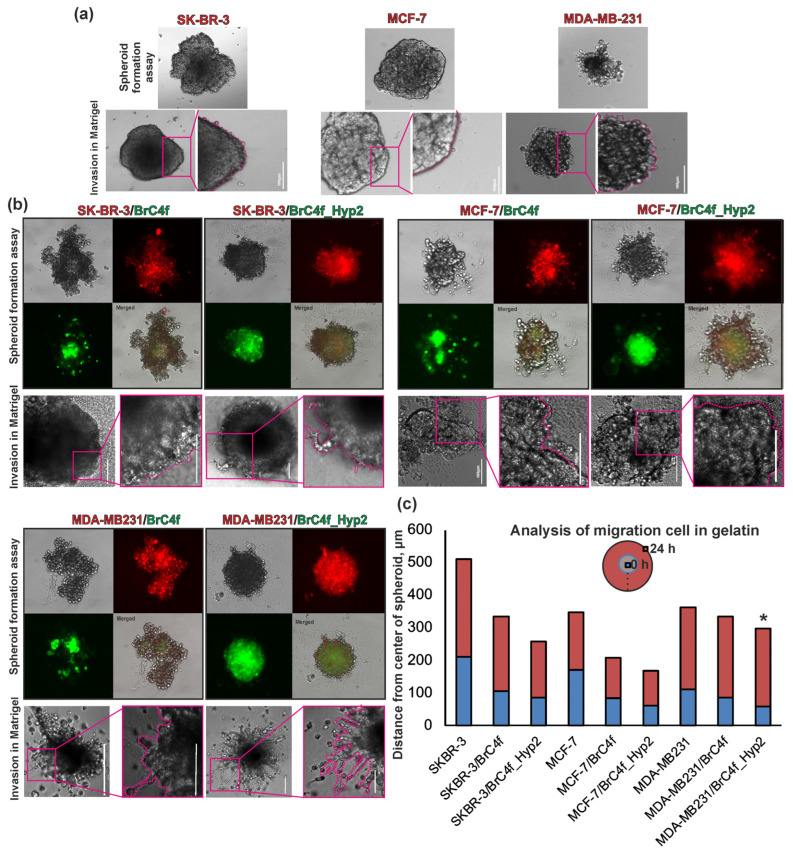
Analysis of the effect of hypoxia-activated CAF (BrC4f_Hyp2) in a heterotypic spheroid model on cell motility. Microscopic analysis of 3D-2 during co-culture: red glow—epithelial cells, green—fibroblast-like cells, invasion test—formation of star-shaped protrusions in the Matrigel. Results of invasion test at magnification 10×: (**a**) monotypic spheroid from only tumor cells; (**b**) heterotypic spheroid from tumor cells with BrC4f or BrC4f_Hyp2; (**c**) histogram showing cell migration from spheroids in gelatin. * *p* < 0.05 determined by unpaired two-tailed Student’s t test.

**Table 1 ijms-26-07789-t001:** Initial characteristics of biological material of human tissue used for patient-derived culture preparation.

Cell Culture	Primary Biological Material	Characteristics Obtained by Histological Analysis of the Primary Tumor *
BC3f	Bl mammae	T2N0M0
BC4f	Bl mammae	T1N0M0
BrC1f	Bl mammae	-
BrC4f	Bl mammae	T2N0M0 [21]
Met-Tem	Bl mammae, docetaxel chemotherapy, radiation therapy	Brain metastasis, chemotherapy trastazumab and pertuzumab
BN120f	Normal tissue	-
NSK1f	Normal tissue (skin)	-

Notes: 1. Bl mammae—breast cancer; TNM—tumor, lymph node, metastasis; 2. * Data were obtained from Sidorov S.V. (Doctor of medical sciences, professor, honored doctor of the Russian Federation, head of department, GBUZ NSO ‘Municipal Clinical Hospital No. 1’, Novosibirsk).

**Table 2 ijms-26-07789-t002:** Classification of different subpopulations of fibroblast-like cells of patient-derived cell cultures according to their surface markers.

Cell Culture							
Marker	BC3f	BC4f	BrC1f	BrC4f	Met-Tem	BN120f	NSK1f
Markers of fibroblast-like cells							
Mel-CAM	low	-	-	high	-	-	low
Ep-CAM	-	-	-	-	-	-	-
E-Cad	-	-	low	-	low	low	-
N-Cad	high	high	high	high	low	high	high
EGFR	high	+	low	+	low	high	high
Vim	+	+	+	+	+	+	+
Ki-67	med	med	high	high	med	-/low	-/low
Markers of cancer-associated fibroblasts						Normal fibroblasts	
αSMA	low	med	low	high	med	low	high
FAPα	med	high	med	med	med	high	high
FSP1/S100A4	med	high	low	high	med	low	high
F-aктин	+	+	+	+	+	+	+
PDGFRα	-	-	low	-	low	-	-
CD90	+	+	+	high	low	+	high
Type of cancer-associated fibroblasts						Normal fibroblasts	
Cell culture							
	BC3f	BC4f	BrC1f	BrC4f	Met-Tem	BN120f	NSK1f
	**S3**	**S1**	**S2**	**S4**	**S1**	-	-

Notes: “-” is negatives or “+” are positive cells population.

**Table 3 ijms-26-07789-t003:** Cytotoxic activity of antitumor drugs against patient-derived BC cell cultures and normal human cells (see the next page).

Cell Culture	IC50		
	Doxorubicin, µM	Cisplatin, µM	Tamoxifen, µM
Cancer-associated fibroblasts			
BC3f	1.19	n/d	68.4
BC4f	0.69	26.15	8.4
BrC1f	n/d	n/d	n/d
BrC4f	0.69	34.48	26.92
Met-Tem	1.33	n/d	n/d
Normal fibroblasts			
BN120f	4.07	n/d	19.46
NSK1f	4.8	22.4	15.6
Immortalized breast cancer cell line			
MCF7	1.16	n/d	49.5
MDA-MB-231	1.06	n/d	n/d

Notes: n/d—the value of cell sensitivity to the drug is not determined in the studied concentration range. MCF7, MDA-MB-231 used as additional controls. IC50—concentration of the drug at which 50% of cells are dead.

**Table 4 ijms-26-07789-t004:** Functional characteristics of CAFs in TME.

Cell Culture							
Type of Cancer-Associated Fibroblasts						Normal Fibroblasts	
	BC3f	BC4f	BrC1f	BrC4f	Met-Tem	BN120f	NSK1f
CAF type	S3	S1	S2	S4	S1	n/a	n/a
Functionality	IS	IS and MM	SC/PC	MM	Met	N	N
Possibility of plasticity	+	+	-	-	+	+	+

Notes: n/a—not applicable; IS—immunosuppressive; MM—matrix-modeling; SC/PC—stem/progenitor; Met—aggressive metastatic; N—normal; “-”—no possibility or “+”—possibility.

## Data Availability

The data that support the findings of this study are available from the authors upon reasonable request.

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
