# Peer review of "Exploring the Heterogeneity of Cancer-Associated Fibroblasts via Development of Patient-Derived Cell Culture of Breast Cancer"

_ijms, 2025, doi:10.3390/ijms26167789_

Round 1

Reviewer 1 Report

Comments and Suggestions for Authors

The authors aim to investigate an important and timely question regarding the role of Cancer-Associated Fibroblasts (CAFs) in breast cancer. To this end, they established primary cell cultures from breast cancer tumors to characterize distinct CAF subtypes and evaluated the cytotoxic effects of several chemotherapeutic agents, including doxorubicin, cisplatin, and tamoxifen, across four CAF populations. They further hypothesized that CAF-derived spheroids could serve as an in vitro model for breast cancer-associated fibrosis.

While the study addresses a conceptually valuable topic, the overall quality of the data and presentation, particularly the imaging, is insufficient for publication. In its current form, the manuscript suffers from poor organization, limited originality, and several unclear conclusions. Notably, key data, such as immunostaining for CAF markers, are fragmented across seven figures without integration or cross-comparison. I strongly recommend consolidating these data and significantly enhancing the imaging quality to support the findings summarized in Table 2.

Specific Concerns:

Abstract: The abstract lacks coherence and fails to present a unified hypothesis. It reads as a list of unrelated observations, mentioning drug cytotoxicity, fibrosis modeling, and hypoxia-induced CAF activity without clearly linking these elements under a central aim or logical progression.

Line 22: The phrase "a transistor spectrum of phenotypes" is unclear and potentially incorrect. If the authors meant a continuum or range, they should revise accordingly.

Line 119: The term "vesicle separation" is ambiguous. Clarification is needed—does this refer to exosome isolation, intracellular trafficking, or another process?

Figure 1: The hematoxylin staining does not visibly mark nuclei, which raises concerns about staining quality or image contrast.

The origin and classification of fibroblasts are confusing. What distinguishes "Met-Tem," "NSK1f," and other subtypes? Which fibroblasts are derived from the eyelid? These distinctions must be clearly defined in the figure legend and methods.

Figure 2: The methodology for measuring adhesion, proliferation, and viability is not described. How are these metrics integrated into a single Cell Index curve? Brightfield images of each fibroblast population at various time points should be included to illustrate morphological characteristics and confluency. A reference fibroblast line (normal and tumor-derived) with known adhesion, proliferation, and viability parameters should be included as a control.

Figure 3: The image resolution is poor—magnified panels are blurry and not suitable for quantitative interpretation.

Vimentin expression varies widely across cultures. Is vimentin used as a marker to confirm fibroblast identity? Additionally, representative images of stained fields should accompany the bar graph for Ki67 in Figure 3a.

Figure 4: FSP1 staining appears irregular and punctate, which is atypical. FSP1 should show a more uniform distribution.

It is unclear how mean fluorescence intensity (MFI) was quantified given the inconsistency in staining. Additional replicates and controls are needed to validate these measurements.

Figures 5–7: The image quality in these figures is unacceptably low, and the conclusions drawn are not supported by the presented data.

Figure 9: This model presents an interesting system to study how hypoxia-activated CAFs (BrC4f_Hyp2) influence breast cancer cell behavior in 3D co-culture. However, major limitations compromise the findings: No molecular analysis was conducted to identify factors responsible for the observed phenotypes. Increases in spheroid size or dispersion could result from cell proliferation, not migration. The hypoxia induction protocol is not described: What oxygen concentration was used? For how long? Was it acute, chronic, or cyclic?

Discussion: The discussion provides some valuable reflections on CAF heterogeneity and the need for improved in vitro models, but it is weakened by structural and scientific shortcomings:

  • There is minimal comparison to relevant literature. While prior studies on hypoxia-induced CAF activity are cited, the authors fail to critically discuss how their findings align or diverge from these reports.
  • The claim that “we were unable to find similar studies” on CAF-derived spheroids suggests an inadequate literature review. Numerous recent studies have successfully generated fibroblast-only spheroids in fibrosis and cancer research.
  • The narrative lacks cohesion, moving abruptly between topics—drug screening, CAF classification, fibrosis modeling, tumor spheroids—without clear transitions.
  • Key themes such as CAF heterogeneity and invasion are repeated without additional depth or synthesis, reducing the impact of the discussion.

In summary, while the study has conceptual merit, it requires substantial revision to meet publication standards. Improved experimental design, enhanced data quality (particularly imaging), clearer presentation of findings, and more rigorous engagement with existing literature are necessary to support the authors’ conclusions.

Author Response

Reviewer 1.

Comments for Reviewer 1:

Point 1. In its current form, the manuscript suffers from poor organization, limited originality, and several unclear conclusions. Notably, key data, such as immunostaining for CAF markers, are fragmented across seven figures without integration or cross-comparison. I strongly recommend consolidating these data and significantly enhancing the imaging quality to support the findings summarized in Table 2.

Response 1: Corrections have been made to the manuscript, with the objective of enhancing the quality of the presentation and ensuring consistency in the description of the results. It is also important to note that the figures were changed for the purpose of enhancing the visual appeal of the work. The differences between cell cultures and separated CAFs and normal have been clearly delineated.

Point 2. Abstract: The abstract lacks coherence and fails to present a unified hypothesis. It reads as a list of unrelated observations, mentioning drug cytotoxicity, fibrosis modeling, and hypoxia-induced CAF activity without clearly linking these elements under a central aim or logical progression.

Response 2: Thank you for your kind consideration of our work. We structured the abstract of the manuscript, which will provide a clear overview of its content. See line 19-36.

Point 3. Line 22: The phrase "a transistor spectrum of phenotypes" is unclear and potentially incorrect. If the authors meant a continuum or range, they should revise accordingly.

Response 3: The phrase was changed to «the plasticity of the CAF phenotype» (See line 26).

Point 4. Line 119: The term "vesicle separation" is ambiguous. Clarification is needed—does this refer to exosome isolation, intracellular trafficking, or another process?

Response 4: This statement has been taken out of the current manuscript.

Point 5. Figure 1: The hematoxylin staining does not visibly mark nuclei, which raises concerns about staining quality or image contrast. The origin and classification of fibroblasts are confusing. What distinguishes "Met-Tem," "NSK1f," and other subtypes? Which fibroblasts are derived from the eyelid? These distinctions must be clearly defined in the figure legend and methods.

Response 5: In current version of manuscript changed Figure 1. Distinctions are defined in the figure legend and methods (See line 147-153 and 674-678 Improvements were made in the characterization of the observed morphology and these were then compared to the preexisting control cell lines (See line 125-144).

Point 6. Figure 2: The methodology for measuring adhesion, proliferation, and viability is not described. How are these metrics integrated into a single Cell Index curve? Brightfield images of each fibroblast population at various time points should be included to illustrate morphological characteristics and confluency. A reference fibroblast line (normal and tumor-derived) with known adhesion, proliferation, and viability parameters should be included as a control.

Response 6: An explanation of the method and data interpretation has been incorporated. In addition, a control system was incorporated (See Fig 2, line 159-175, 747-756). Interpretation of xCELLigence Cell Index curves: The strength of cell adhesion is represented by the software as the Cell Index (unit-less measurement). As cells adhere to the E-plates, the Cell Index value increases from zero, and this will usually be evident within the first 10–15 min of seeding 10.3390/bios5020199 10.1039/D0TB01454K 10.3892/etm.2017.4781. In instances wherein the utilisation of xCELLigence is employed, there becomes an unnecessary requirement for the generation of brightfield images. The functionality intrinsic to such apparatus is predicated upon its capacity to operate in a real-time capacity.

Point 7. Figure 3: The image resolution is poor—magnified panels are blurry and not suitable for quantitative interpretation.

Response 7: All figures have been corrected and re-uploaded to make them more visually comparable and easier to understand.

Point 8. Vimentin expression varies widely across cultures. Is vimentin used as a marker to confirm fibroblast identity? Additionally, representative images of stained fields should accompany the bar graph for Ki67 in Figure 3a.

Response 8: Vimentin's ubiquitous expression in the whole fibroblast population as well as multiple other cell types, including macrophages, adipocytes, and the cells undergoing EMT severely limits its utility as a CAF-specific marker. We added this point to the manuscript (See line 197-201). Ki-67 staining (red signal) was performed in conjunction with vimentin staining (See Fig 3).

Point 9. Figure 4: FSP1 staining appears irregular and punctate, which is atypical. FSP1 should show a more uniform distribution.

Response 9: Thank you for your important observation. Classical FSP1 (S100A4) expression in fibroblasts is typically characterized by cytoplasmic and/or nuclear distribution 10.3389/fimmu.2025.1525567. We hypothesize that this is a biological feature, including the possible presence of a secreted form of FSP1 in BC4f and BrC4f. FSP1 secretion and association with components of the extracellular matrix (e.g. fibronectin) or neighboring cell receptors will inevitably result in heterogeneous staining around stromal cells 10.1038/s41598-023-49863-y. We added this point to the manuscript (See line 284-286, 294-297). Furthermore, this atypical staining is indicative of a single previously characterized BrC4f cell line. This particular cell culture is distinguished by its ability to undergo a mesenchymal-epithelial transition when exposed to specific conditions 10.3390/ijms24032494. Research conducted by 10.2174/156652408785747942 and 10.2353/ajpath.2010.090526 has indicated that extracellular FSP1 may be implicated in the following processes of fibroblast activation and matrix remodelling. BC4f and BrC4f cell cultures is characterized as matrix-modeling by a set of markers (See Table 2,4).

Point 10. It is unclear how mean fluorescence intensity (MFI) was quantified given the inconsistency in staining. Additional replicates and controls are needed to validate these measurements.

Response 10: We have now added some information on how MFI is calculated to the Materials and Methods section (See line 786-827). We also used data from epithelial-like MCF-7 and mesenchymal-like MDA-MB-231 breast cancer cell lines as a control.

Point 11. Figures 5–7: The image quality in these figures is unacceptably low, and the conclusions drawn are not supported by the presented data.

Response 11: All figures have been corrected and re-uploaded to make them more visually comparable and easier to understand.

Point 12. Figure 9: This model presents an interesting system to study how hypoxia-activated CAFs (BrC4f_Hyp2) influence breast cancer cell behavior in 3D co-culture. However, major limitations compromise the findings: No molecular analysis was conducted to identify factors responsible for the observed phenotypes. Increases in spheroid size or dispersion could result from cell proliferation, not migration. The hypoxia induction protocol is not described: What oxygen concentration was used? For how long? Was it acute, chronic, or cyclic?

Response 12: The present study utilized a previously developed approach to modelling hypoxia conditions, termed «pulsed hypoxia» doi.org/10.3390/ijms24032494. «Pulsed hypoxia» is based on the repetitive switching of the cultivating condition from normoxia to hypoxia. The ratio of oxygen portion in the medium during “pulsed hypoxia” changes by 7% doi.org/10.3390/ijms24032494. The hypoxia induction protocol and oxygen concentration are described in section Material and methods (See line 393-395, 722-729). As outlined in our previous work, which is referenced in the current manuscript, the molecular and cellular changes that occur in cells under hypoxia are described in detail doi.org/10.3390/ijms24032494.

We concur with the reviewer's assertion that proliferation has the capacity to influence the dimensions and distribution of spheroids within Matrigel. Indeed, an increase in the size or dispersion of spheroids in Matrigel can be attributed to both cell migration and proliferation. In the present experiment, Matrigel was utilized as a medium that emulates the extracellular matrix, thereby engendering conditions conducive to migration. Gelatin is often used as a substrate in cell migration assays to study cell movement and invasion 10.3389/fcell.2019.00107. In order to mitigate the impact of proliferation, experiments were conducted over a limited time period. However, the study's primary focus did not pertain to spheroid change, but rather to the examination of cell exiting the spheroid. This is a crucial component in the assessment of the potential for migration and invasion. The current version of the manuscript incorporates additional controls from monotypic spheroids derived exclusively from tumor cells (See Fig. 6a). As demonstrated in the Figure 6, there is an absence of discernible invasion processes under the experimental conditions outlined.

Point 13. Discussion: The discussion provides some valuable reflections on CAF heterogeneity and the need for improved in vitro models, but it is weakened by structural and scientific shortcomings.

Response 13: We would like to express our gratitude to the reviewer for their constructive feedback, which undoubtedly contribute to the enhancement of our manuscript.

Point 14. There is minimal comparison to relevant literature. While prior studies on hypoxia-induced CAF activity are cited, the authors fail to critically discuss how their findings align or diverge from these reports.

Response 14: The discussion section has been expanded to facilitate in-depth discourse regarding our results (See line 500-559, 608-611, 627-638).

Point 15. The claim that “we were unable to find similar studies” on CAF-derived spheroids suggests an inadequate literature review. Numerous recent studies have successfully generated fibroblast-only spheroids in fibrosis and cancer research.

Response 15: The aforementioned statement was removed and replaced with a series of studies focusing on the acquisition of stromal spheroidal models (See lines 579-594).

Point 16. The narrative lacks cohesion, moving abruptly between topics—drug screening, CAF classification, fibrosis modeling, tumor spheroids—without clear transitions.

Key themes such as CAF heterogeneity and invasion are repeated without additional depth or synthesis, reducing the impact of the discussion.

Response 16: We added clear transitions in the description of the results and their discussion in the manuscript. (See Results and Discussion section)

Point 17. In summary, while the study has conceptual merit, it requires substantial revision to meet publication standards. Improved experimental design, enhanced data quality (particularly imaging), clearer presentation of findings, and more rigorous engagement with existing literature are necessary to support the authors’ conclusions.

Response 17: We sincerely appreciate the time and effort the reviewer has dedicated to evaluating our manuscript. We are grateful for the constructive feedback, which has helped us identify key areas for improvement. In the current version of the manuscript, we corrected minor spelling and typo errors. Current version of the manuscript was proofed by English speaker.

Reviewer 2 Report

Comments and Suggestions for Authors

This study systematically explored the heterogeneity of cancer-associated fibroblasts (CAFs) and their functions in the tumor microenvironment by establishing a primary stromal cell model of breast cancer. The authors successfully isolated and characterized multiple CAF subtypes (S1-S4), revealing differences in their phenotypic plasticity and drug sensitivity. It was found that hypoxia-activated CAF-S4 can significantly promote the migration and invasion of triple-negative breast cancer cells, providing new ideas for therapies targeting CAFs. Combining 3D spheroid models, multi-omics marker analysis and functional experiments, the tumor fibrosis microenvironment was simulated more comprehensively. However, there are deviations between the CAFs classification criteria and the Costa scheme (such as normal fibroblasts are included in the classification), and key conclusions (such as CAF-S2 drug resistance) lack mechanistic support.
1. The introduction part supplements the current status of the controversy over the classification of CAFs heterogeneity (such as the difference between single-cell sequencing and protein expression) and clarifies the reasons for adopting the Costa classification.
2. Verification of CAFs classification (Figures 1-7, Table 2): Unified classification standards: Normal fibroblasts (BN120f, NSK1f) should not be included in CAFs subtypes (Table 2). Additional explanation: "Normal tissue fibroblasts are only used as controls and are not involved in S1-S4 classification."
3. Error bars should be added to the bar graphs of Figures 3a, 4b, and 5b, and statistical methods (such as SD/SEM, n value) should be indicated.
4. Supplement clues to drug resistance mechanisms: such as CAF-S2 highly expressing drug resistance genes (ABC transporters).
5. Emphasize the innovativeness of "single CAF spheres" (non-mixed tumor cells) to simulate fibrotic lesions.
6. Quantification of invasion/migration: Figure 9d needs to add Y-axis units (migration distance μm) and annotate significant differences (such as ***p<0.001).
7. Associate the metastatic effect of hypoxic CAF-S4 with specific secreted factors (such as LOX, MMPs). Explain whether CAF-S2 resistance is related to stem cell characteristics ("SC/PC" in Table 2).
8. Combine the annotations of Figure 9a-c sub-figures (SKBR3/MCF7/MDA-MB-231) to avoid duplication; supplement the control group (no CAF) data in Figure d.
9. The abbreviations in the whole article need to be defined for the first time (such as TME, ECM, scRNA-Seq).

Author Response

Reviewer 2.

We thank the reviewer for the positive feedback on our manuscript.

Point 1. However, there are deviations between the CAFs classification criteria and the Costa scheme (such as normal fibroblasts are included in the classification), and key conclusions (such as CAF-S2 drug resistance) lack mechanistic support.

Response 1: The manuscript was extensively reviewed and revised, resulting in an updated and refined classification that excludes the classification of normal fibroblasts. Normal fibroblasts from breast (BN120f) and eyelid (NSK1f) tissue were utilized exclusively as controls and did not contribute to S1-S4 classification (See Table 2, 4).

Point 2. The introduction part supplements the current status of the controversy over the classification of CAFs heterogeneity (such as the difference between single-cell sequencing and protein expression) and clarifies the reasons for adopting the Costa classification.

Response 2: In the current version of the manuscript, we have added a discussion about the different methodologies used to classify CAFs (See line 58-69).
Point 3.  Verification of CAFs classification (Figures 1-7, Table 2): Unified classification standards: Normal fibroblasts (BN120f, NSK1f) should not be included in CAFs subtypes (Table 2). Additional explanation: "Normal tissue fibroblasts are only used as controls and are not involved in S1-S4 classification."

Response 3: In current version of manuscript all figures have undergone modification for the purpose of enhancing the visual appeal of the work. A clear delineation was provided between the differences observed in cell cultures and isolated CAFs in comparison to normal cells. Control samples comprising epithelial-like MCF-7 and mesenchymal-like MDA-MB-231 breast cancer cell lines have been incorporated into all experiments.
Point 4.   Error bars should be added to the bar graphs of Figures 3a, 4b, and 5b, and statistical methods (such as SD/SEM, n value) should be indicated.

Response 4: We have added data on MFI. Data are mean ± standard deviation of three replicate measurements performed on the same cells.
Point 5.    Supplement clues to drug resistance mechanisms: such as CAF-S2 highly expressing drug resistance genes (ABC transporters).

Response 5: A discussion of this cellular functionality has now been appended to the manuscript (See line 500-560).
Point 6.     Emphasize the innovativeness of "single CAF spheres" (non-mixed tumor cells) to simulate fibrotic lesions.

Response 6: We added this point to the manuscript (See line 390-392).
Point 7.    Quantification of invasion/migration: Figure 9d needs to add Y-axis units (migration distance μm) and annotate significant differences (such as ***p<0.001).

Response 7: The Figure and its caption have been corrected (See Fig. 6).
Point 8.    Associate the metastatic effect of hypoxic CAF-S4 with specific secreted factors (such as LOX, MMPs). Explain whether CAF-S2 resistance is related to stem cell characteristics ("SC/PC" in Table 2).

Response 8: A discussion of this cellular functionality has now been appended to the manuscript (See line 608-611, 627-638).
Point 9.     Combine the annotations of Figure 9a-c sub-figures (SKBR3/MCF7/MDA-MB-231) to avoid duplication; supplement the control group (no CAF) data in Figure d.

Response 9: The Figure and its caption have been corrected (See Fig. 6). In the present study, results for monotypic spheroids have been added to all tests (See Fig6a,c).
Point 10.     The abbreviations in the whole article need to be defined for the first time (such as TME, ECM, scRNA-Seq).

Response 10: The full list of abbreviations has been appended as additional material (See line 962).

Round 2

Reviewer 1 Report

Comments and Suggestions for Authors

While the study aims to investigate the phenotypic and functional diversity of cancer-associated fibroblasts (CAFs) in breast cancer, the manuscript suffers from substantial conceptual, methodological, and presentation issues that preclude its acceptance in its current form. After the first revision, I still think there is a lack of novelty and scientific rigor in foundational results, as well as inconsistency and poor validation across assays. There are still persistent issues with image quality and presentation, as well as fundamental problems with manuscript organization and clarity.

I explain below the major issues I found in the revised manuscript, focusing on the results:

Result 2.1: The observation of cellular heterogeneity across patient-derived cultures (BC3f, BC4f, BrC1f, Met-Tem) is expected and, therefore, not particularly insightful. The characterization provided in Figure 1 relies heavily on vague morphological descriptors such as "visible nucleus" or "dense cytoplasm," which lack quantifiable or diagnostic relevance. The rationale behind including H&E staining is questionable, especially when hematoxylin staining is incomplete or absent, and the cellular features described offer limited scientific value.

The analysis of cell index data is also problematic. The legend and curve labeling are not aligned; the order of cell lines on the graph should match the sequence in the legend for clarity. Notably, the finding that Met-Tem cells show a highly proliferative profile, while MCF7 and MDA-MB-231 are less so, may be biologically meaningful. However, it is puzzling that BrC4f—a commonly used control—is shown to be non-proliferative. This raises concerns about either the biological validity of the data or potential inconsistencies in how the assay was conducted or interpreted.

Result 2.2: This paragraph is mistakenly numbered 3.2

The Ki67 staining shows different proliferation rates than the cell index above. How do the authors explain this?

Ki 67: 1%_≤NSK1f<BN120f<BC4f<Met-Tem≤BC3f<BrC4f<BrC1f≤50%≤MDA-MB-231≤MCF7≤80%

Cell Index: <BC4f%≤MDA-MB-231≤MCF7<BN120f<BrC1f≤NSK1f≤BC3f< BC4f<Met-Tem

The images presented in Figure 3 for Ki67 staining are still of poor quality.

The authors used several markers like vimentin, CD90, EMT markers, PDGFR, and CD44, attributing multiple different characteristics to the CAFs cultures they established, but their rationale is very confusing.

Result 2.3 This section reflects the most substantive improvement in the revised manuscript. Side-by-side comparison of multiple fibroblast markers strengthens the characterization of CAF cultures. However, the immunofluorescence images remain inadequate. In cases where dual labeling is used (e.g., FSP1 and F-actin or EGFR and αSMA), it is unclear which signal corresponds to which marker. Proper labeling, channel separation, and higher-resolution images are essential for drawing valid conclusions.

Result 2.5: Potential for clustering of fibroblasts into spheroids

The observation that CAFs may form spheroids or organoid-like structures is interesting, but the presentation is not suitable for inclusion in the main figures. The brightfield and GFP images the authors present are only suitable for supplemental material.

Result 2.6: Hypoxia-activated CAFs on breast tumor cells

This final section remains poorly structured and difficult to follow. The authors appear to be testing the hypothesis that hypoxia-activated CAFs enhance both the invasiveness and migratory behavior of breast cancer cells, particularly in triple-negative subtypes. However, the data presentation is fragmented, and key conclusions are obscured by excessive methodological detail that should be relocated to the methods section.

Author Response

Comments for Reviewer 1:

Point 1. While the study aims to investigate the phenotypic and functional diversity of cancer-associated fibroblasts (CAFs) in breast cancer, the manuscript suffers from substantial conceptual, methodological, and presentation issues that preclude its acceptance in its current form. After the first revision, I still think there is a lack of novelty and scientific rigor in foundational results, as well as inconsistency and poor validation across assays. There are still persistent issues with image quality and presentation, as well as fundamental problems with manuscript organization and clarity.

Response 1: Thank you for your careful consideration of our work and for your valuable comments. Having analyzed your comments, we are afraid that we cannot agree with all of them. We agree that a large body of literature exists on the generation of new cell cultures, including cancer-associated fibroblasts (CAFs). Moreover, the approaches to studying CAFs can be considered to comprise the following four interconnected strategies: (1) isolation via primary culture or cytokine-induced differentiation; (2) animal models, such as genetically engineered mouse models; (3) in vitro systems, such as 2D/3D co-culture, conditioned medium analysis and organoids, or in vivo co-implantation, to analyze tumor-CAF interactions at cellular and molecular levels; Fourthly, combinations of classical techniques (such as immunohistochemistry and flow cytometry) are used alongside emerging tools (such as single-cell RNA sequencing (scRNA-seq) and spatial transcriptomics) to reveal the complex molecular functions of CAFs within the tumor microenvironment. We eddied this point in introduction (See line 67-76, 88-91). Additionally, the majority of ongoing preclinical trials focus on the entire CAF population, disregarding their heterogeneous nature and multifaceted functions within tumors. Although a wealth of information has been brought by studies attempting to classify CAFs based on scRNA-seq, there are some limitations in these studies. scRNA-seq analyses can be expensive and time-consuming, but they provide a snapshot of the entire transcriptome of individual cells and gene expression patterns. However, they do not provide information on proteins. Nevertheless, single-cell platforms for spatial transcriptomics have ultra-high throughput, clustering and detection of CAF transcriptome states may be limited by gene coverage and lack of key markers. Immunocytochemistry and flow cytometry are used to detect and quantify specific proteins on or within cells. To the best of our knowledge, there are few other publications with similar collections of fibroblasts characterized by protein markers and functional tests using one of the most well-known classifications according to CAF types.

We recognize that some aspects of the validation process were not set out in sufficient detail in the original version of the manuscript, and we have endeavored to address this in the revised version. Differences in data between experiments may reflect the biological variability of CAFs under different conditions, which is an important aspect of their heterogeneity. Nevertheless, we have analyzed possible sources of discrepancy in more detail and clarified their interpretation in the text.

Point 2. Result 2.1: The observation of cellular heterogeneity across patient-derived cultures (BC3f, BC4f, BrC1f, Met-Tem) is expected and, therefore, not particularly insightful. The characterization provided in Figure 1 relies heavily on vague morphological descriptors such as "visible nucleus" or "dense cytoplasm," which lack quantifiable or diagnostic relevance. The rationale behind including H&E staining is questionable, especially when hematoxylin staining is incomplete or absent, and the cellular features described offer limited scientific value.

Response 2: We repeated the experiment using hematoxylin-eosin staining on the cells, and transferred the Figure S1 to the supplementary data (See Fig.S1). The discussion of the Figure S1 was removed, reducing it to a single sentence (See line 138-139).

Point 3. The analysis of cell index data is also problematic. The legend and curve labeling are not aligned; the order of cell lines on the graph should match the sequence in the legend for clarity. Notably, the finding that Met-Tem cells show a highly proliferative profile, while MCF7 and MDA-MB-231 are less so, may be biologically meaningful. However, it is puzzling that BrC4f—a commonly used control—is shown to be non-proliferative. This raises concerns about either the biological validity of the data or potential inconsistencies in how the assay was conducted or interpreted.

Response 3: A re-analysis of the proliferation assay for BrC4f cells was conducted. The updated data have now been incorporated (see lines 153-156). The resulting CI does not represent a direct measurement of cell size and typical proliferation curves, but rather reflects the overall status of the cell layer within the well. Therefore, while the cell index is influenced by cell size, it is also contingent on other factors, including cell morphology, adhesion strength, and cell density. It is important to note that spreading can alter the CI. The increase of CI is a result of cells proliferating, adhering strongly and spreading out on the electrode surface. Strongly adherent cells will have a higher CI than weakly adherent cells, which is specific to stromal cells. We added this point in current version of manuscript (See line 147-149).

Point 4. Result 2.2: This paragraph is mistakenly numbered 3.2

Response 4: Corrected the typo.

Point 5. The Ki67 staining shows different proliferation rates than the cell index above. How do the authors explain this?

Ki67:1%_≤NSK1f<BN120f<BC4f<MetTem≤BC3f<BrC4f<BrC1f≤50%≤MDAMB231≤MCF7≤80%

Cell Index: <BC4f%≤MDA-MB-231≤MCF7<BN120f<BrC1f≤NSK1f≤BC3f< BC4f<Met-Tem

The images presented in Figure 3 for Ki67 staining are still of poor quality.

Response 5: Ki67 is a marker of the proportion of cells in the active cell cycle (G1, S, G2, M phases); however, it does not reflect changes in proliferation dynamics over time. The real-time cell index is a valuable tool that can be used to monitor changes in adhesion, morphology and proliferation. These changes are influenced by various factors, including the division rate, metabolic activity, differentiation and cell death. The CI is susceptible to the influence of the extracellular matrix, the seeding density, and the culture conditions. These factors do not necessarily correlate with Ki67 levels.

The proliferation of normal fibroblasts is characterized by a low rate of growth, which is reflected in the low Ki-67 levels observed in the data. https://doi.org/10.1242/jcs.107.2.571. Tumor epithelial cells as (MCF7 or MDA-MB-231) have high levels of Ki-67. 10.3390/cancers13174455

CAFs is group comprises native, normal fibroblasts (low Ki-67), as well as activated, proliferating (Ki67+) or recruited fibroblasts in response to cancer-derived stimuli. Ki67 expression levels may be reduced as a consequence of the activity of non-proliferative functions of CAF (e.g. invasive potential (Met-Tem) or matrix remodelling (BrC4f)). CAFs exhibited a moderately elevated Ki-67 index, though typically at levels lower than those observed in epithelial tumour cells doi.org/10.3389/fimmu.2025.1582532 doi.org/10.1038/s41467-023-39762-1 We added it in discussion (See line 472-474, 502-506).

The latest guidelines were adhered to during the processes of sample preparation and counting, for example doi.org/10.3390/curroncol30030233. In accordance with the publisher's recommendations, figure resolutions have been incorporated.

Point 6. The authors used several markers like vimentin, CD90, EMT markers, PDGFR, and CD44, attributing multiple different characteristics to the CAFs cultures they established, but their rationale is very confusing.

Response 6: The choice of these markers was based on their well-recognized role in the identification and characterization of fibroblast and their association with functional heterogeneity of the breast tumor stroma. Importantly, no specific markers of fibroblasts have been found so far that are not expressed in other cell types. We have included an explanation of why these markers were chosen for the study (See line 188-196, 206-207, 211-228, 238-242, 475-480).

Point 7. Result 2.3 This section reflects the most substantive improvement in the revised manuscript. Side-by-side comparison of multiple fibroblast markers strengthens the characterization of CAF cultures. However, the immunofluorescence images remain inadequate. In cases where dual labeling is used (e.g., FSP1 and F-actin or EGFR and αSMA), it is unclear which signal corresponds to which marker. Proper labeling, channel separation, and higher-resolution images are essential for drawing valid conclusions.

Response 7: Existing images are divided into channels to improve visualization. The figure caption has been updated to include protein labelling. We would also like to point out that, when calculating MFI in ImageJ, the analysis is performed on each channel rather than using co-labelling. The resolution of the figures is given in the manuscript in accordance with the publisher's recommendations.

Point 8. Result 2.5: Potential for clustering of fibroblasts into spheroids

The observation that CAFs may form spheroids or organoid-like structures is interesting, but the presentation is not suitable for inclusion in the main figures. The brightfield and GFP images the authors present are only suitable for supplemental material.

Response 8: The recommendation was followed and the brightfield images of spheroids (See Figure S4) were removed from the main figures. These are not images of GFP, but analyses of cell viability as part of a 3D model by staining FDA/PI/Hoechst. This image was included in the main body of the manuscript. We believe this to be an important finding, given that previous studies have shown that stromal cells are subject to nemosis under such culturing conditions.

Point 9. Result 2.6: Hypoxia-activated CAFs on breast tumor cells

This final section remains poorly structured and difficult to follow. The authors appear to be testing the hypothesis that hypoxia-activated CAFs enhance both the invasiveness and migratory behavior of breast cancer cells, particularly in triple-negative subtypes. However, the data presentation is fragmented, and key conclusions are obscured by excessive methodological detail that should be relocated to the methods section.

Response 9: The quality of the presentation of the text has been enhanced through a focus on the results, with the section on the materials and methods being streamlined to exclude methodological explanations.

Reviewer 2 Report

Comments and Suggestions for Authors

The author has responded to the reviewer's concerns and the manuscript can be accepted.

Comments on the Quality of English Language

The author has responded to the reviewer's concerns and the manuscript can be accepted.

Author Response

We thank the reviewer for the positive feedback on our manuscript.

Round 3

Reviewer 1 Report

Comments and Suggestions for Authors

I have reviewed the revised manuscript and find the authors’ responses and revisions satisfactory. I recommend the manuscript for publication in its current form.